# Graph2Seq: Scalable Learning Dynamics for Graphs

## Abstract

Neural networks have been shown to be an effective tool for learning algorithms over graph-structured data. However, *graph representation* techniques—that convert graphs to real-valued vectors for use with neural networks—are still in their infancy. Recent works have proposed several approaches (e.g., graph convolutional networks), but these methods have difficulty scaling and generalizing to graphs with different sizes and shapes. We present Graph2Seq, a new technique that represents vertices of graphs as infinite time-series. By not limiting the representation to a fixed dimension, Graph2Seq scales naturally to graphs of arbitrary sizes and shapes. Graph2Seq is also reversible, allowing full recovery of the graph structure from the sequences. By analyzing a formal computational model for graph representation, we show that an unbounded sequence is necessary for scalability. Our experimental results with Graph2Seq show strong generalization and new state-of-the-art performance on a variety of graph combinatorial optimization problems.

## 1 Introduction

Graph algorithms appear in a wide variety of fields and applications, from the study of gene interactions (Özgür et al., 2008) to social networks (Ugander et al., 2011) to computer systems (Grandl et al., 2016). Today, most graph algorithms are designed by human experts. However, in many applications, designing graph algorithms with strong performance guarantees is very challenging. These algorithms often involve difficult combinatorial optimization problems for which finding optimal solutions is computationally intractable or current algorithmic understanding is limited (e.g., approximation gaps in CS theory literature).

In recent years, deep learning has achieved impressive results on many tasks, from object recognition to language translation to learning complex heuristics directly from data (Silver et al., 2016; Krizhevsky et al., 2012). It is thus natural to ask whether we can apply deep learning to automatically learn complex graph algorithms. To apply deep learning to such problems, graph-structured data first needs to be *embedded* in a high dimensional Euclidean space. *Graph representation* refers to the problem of embedding graphs or their vertices/edges in Euclidean spaces. Recent works have proposed several graph representation techniques, notably, a family of representations called graph convolutional neural networks (GCNN) that use architectures inspired by CNNs for images (Bruna et al., 2014; Monti et al., 2017; Khalil et al., 2017; Niepert et al., 2016; Defferrard et al., 2016; Hamilton et al., 2017b; Bronstein et al., 2017; Bruna & Li, 2017). Some GCNN representations capture signals on a fixed graph while others support varying sized graphs.

In this paper, we consider how to learn graph algorithms in a way that generalizes to large graphs and graphs of different topologies. We ask: *Can a graph neural network trained at a certain scale perform well on orders-of magnitude larger graphs (e.g.,* $1000\times$ *the size of training graphs) from diverse topologies?* We particularly focus on learning algorithms for combinatorial optimization on graphs. Graph neural networks that generalize to large unseen graphs can be useful for many practical applications. For example, in emerging applications of reinforcement learning (RL) in computer systems (e.g., task device placement (Mirhoseini et al., 2017), database query optimization (Krishnan et al., 2018), job scheduling (Mao et al.)), it is often difficult to train RL agents on large graphs, as executing a policy for a large graph (e.g., running a large database query) can be slow. It is therefore desirable to train on small graphs in a way that generalizes to large graphs. Prior works have typically

considered applications where the training and test graphs are similar in scale, shape, and attributes, and consequently have not addressed this generalization problem. For example, Khalil et al. (2017) train models on graph of size 50-100 and test on graphs of up to size 1200 from the same family; Hamilton et al. (2017a) propose inductive learning methods where they train on graphs with $15k$ edges, and test on graphs with $35k$ edges (less than $3\times$ generalization in size); Ying et al. (2018) train and test over the same large Pinterest graph (3 billion nodes).

We propose GRAPH2SEQ, a *scalable* embedding that represents vertices of a graph as a *time-series* (§3). Our key insight is that the fixed-sized vector representation produced by prior GCNN designs limits scalability. Instead, GRAPH2SEQ uses the entire time-series of vectors produced by graph convolution layers as the vertex representation. This approach has two benefits: (1) it can capture subgraphs of increasing diameter around each vertex as the timeseries evolves; (2) it allows us to vary the dimension of the vertex representation based on the input graph; for example, we can use a small number of graph convolutions during training with small graphs and perform more convolutions at test time for larger graphs. We show both theoretically and empirically that this time-series representation significantly improves the scalability and generalization of the model. Our framework is general and can be applied to various existing GCNN architectures.

We prove that GRAPH2SEQ is information-theoretically lossless, i.e., the graph can be fully recovered from the time-series representations of its vertices (§3.1). Our proof leverages mathematical connections between GRAPH2SEQ and causal inference theory (Granger, 1980; Rahimzamani & Kannan, 2016; Quinn et al., 2015). Further, we show that GRAPH2SEQ and many previous GCNN variants are all examples of a certain computational model over graphs that we call LOCAL-GATHER, providing for a conceptual and algorithmic unification. Using this computational model, we prove that unlike GRAPH2SEQ, fixed-length representations fundamentally cannot compute certain functions over graphs.

To apply GRAPH2SEQ, we combine graph convolutions with an appropriate RNN that processes the time-series representations (§4). We use this neural network model, G2S-RNN, to tackle three classical combinatorial optimization problems of varying difficulty using reinforcement learning: *minimum vertex cover, maximum cut* and *maximum independent set* (§5). Our experiments show that GRAPH2SEQ performs as well or better than the best non-learning heuristic on all three problems and exhibits significantly better scalability and generalization than previous state-of-the-art GCNN (Khalil et al., 2017; Hamilton et al., 2017a; Lei et al., 2017) or graph kernel based (Shervashidze et al., 2011) representations. Highlights of our experimental findings include:

1. G2S-RNN models trained on graphs of size 15–20 scale to graphs of size 3,200 and beyond. To conduct experiments in a reasonable time, we used graphs of size up to 3,200 in most experiments. However, stress tests show similar scalability even at 25,000 vertices.
2. G2S-RNN models trained on one graph type (e.g., Erdos-Renyi) generalize to other graph types (e.g., random regular and bipartite graphs).
3. G2S-RNN exhibits strong scalability and generalization in each of minimum vertex cover, maximum cut and maximum independent set problems.
4. Training over a carefully chosen adversarially set of graph examples further boosts G2S-RNN's scalability and generalization capabilities.

## 2 RELATED WORK

**Neural networks on graphs.** Early works to apply neural-network-based learning to graphs are Gori et al. (2005); Scarselli et al. (2009), which consider an information diffusion mechanism. The notion of convolutional networks for graphs as a generalization of classical convolutional networks for images was introduced by Bruna et al. (2014). A key contribution of this work is the definition of graph convolution in the *spectral domain* using graph Fourier transform theory. Subsequent works have developed local spectral convolution techniques that are easier to compute (Defferrard et al., 2016; Kipf & Welling, 2016). Spectral approaches do not generalize readily to different graphs due to their reliance on the particular Fourier basis on which they were trained. To address this limitation, recent works have considered *spatial convolution* methods (Khalil et al., 2017; Monti et al., 2017; Niepert et al., 2016; Such et al., 2017; Duvenaud et al., 2015; Atwood & Towsley, 2016). Li et al. (2015); Johnson (2016) propose a variant that uses gated recurrent units to perform the state updates, which has some similarity to our representation dynamics; however, the sequence length is

fixed between training and testing. Veličković et al. (2017); Hamilton et al. (2017a) use additional aggregation methods such as vertex attention or pooling mechanisms to summarize neighborhood states. In Appendix A we show that local spectral GCNNs and spatial GCNNs are mathematically equivalent, providing a unifying view of the variety of GCNN representations in the literature.

Another line of work (Jain et al., 2016; Marcheggiani & Titov, 2017; Tai et al., 2015) combines graph neural networks with RNN modules. They are not related to our approach, since in these cases the sequence (e.g., time-series of object relationship graphs from a video) is already given as part of the input. In contrast our approach generates a sequence as the desired embedding from a single input graph. Perozzi et al. (2014); Grover & Leskovec (2016) use random walks to learn vertex representations in an unsupervised or semi-supervised fashion. However they consider prediction or classification tasks over a fixed graph.

**Combinatorial optimization.** Using neural networks for combinatorial optimization problems dates back to the work of Hopfield & Tank (1985) and has received considerable attention in the deep learning community in recent years. Vinyals et al. (2015); Bello et al. (2016); Kool & Welling (2018) consider the traveling salesman problem using reinforcement learning. These papers consider two-dimensional coordinates for vertices (e.g. cities on a map), without any explicit graph structure. Graves et al. (2016) propose a more general approach: a differential neural computer that is able to perform tasks like finding the shortest path in a graph. The work of Khalil et al. (2017) is closest to ours. It applies a spatial GCNN representation in a reinforcement learning framework to solve combinatorial optimization problems such as minimum vertex cover.

# 3 GRAPHS AS DYNAMICAL SYSTEMS

## 3.1 THE GRAPH2SEQ REPRESENTATION

The key idea behind GRAPH2SEQ is to represent vertices of a graph by the trajectory of an appropriately chosen dynamical system induced by the graph. Such a representation has the advantage of progressively capturing more and more information about a vertex as the trajectory unfolds. Consider a directed graph $G(V, E)$ whose vertices we want to represent (undirected graphs will be represented by having bi-directional edges between pairs of connected vertices). We create a discrete-time dynamical system in which vertex $v$ has a state of $\mathbf{x}_v(t) \in \mathbb{R}^d$ at time $t \in \mathbb{N}$, for all $v \in V$, and $d$ is the dimension of the state space. In GRAPH2SEQ, we consider an evolution rule of the form

$$\mathbf{x}_v(t+1) = \text{relu}(\mathbf{W}_1(\sum_{u \in \Gamma(v)} \mathbf{x}_u(t)) + \mathbf{b}_1) + \mathbf{n}_v(t+1), \tag{1}$$

$\forall v \in V, t \in \mathbb{N}$, where $\mathbf{W}_1 \in \mathbb{R}^{d \times d}$, $\mathbf{b}_1 \in \mathbb{R}^{d \times 1}$ are trainable parameters and $\text{relu}(x) = \max(x, 0)$. $\mathbf{n}_v(\cdot)$ is a $d$-dimensional $(\mathbf{0}, \sigma^2 \mathbf{I})$ Gaussian noise, and $u \in \Gamma(v)$ if there is an edge from $u$ to $v$ in the graph $G$. For any $v \in V$, starting with an initial value for $\mathbf{x}_v(0)$ (e.g., random or all zero) this equation defines a dynamical system, the (random) trajectory of which is the GRAPH2SEQ representation of $v$. More generally, graphs could have features on vertices or edges (e.g., weights on vertices), which can be included in the evolution rule as additional terms within the ReLU function (Appendix B.1); these generalizations are outside the scope of this paper. We use GRAPH2SEQ $(G)$ to mean the set of all GRAPH2SEQ vertex representations of $G$.

**GRAPH2SEQ is invertible.** Our first key result is that GRAPH2SEQ's representation allows recovery of the adjacency matrix of the graph with arbitrarily high probability. Here the randomness is with respect to the noise term in GRAPH2SEQ; see equation 1. In Appendix B.2, we prove:

**Theorem 1.** *For any directed graph $G$ and associated (random) representation GRAPH2SEQ $(G)$ with sequence length $t$, there exists an inference procedure (with time complexity polynomial in $t$) that produces an estimate $\hat{G}_t$ such that $\lim_{t \to \infty} \mathbb{P}[G \neq \hat{G}_t] = 0$.*

Note that there are many ways to represent a graph that are lossless. For example, we can simply output the adjacency matrix row by row. However such representations depend on assigning *labels* or identifiers to vertices, which would cause downstream deep learning algorithms to memorize the label structure and not generalize to other graphs. GRAPH2SEQ's key property is that it is does not depend on a labeling of the graph. Theorem 1 is particularly significant (despite the representation being infinite dimensional) since it shows lossless label-independent vertex representations are possible.

To our understanding this is the first result making such a connection. Next, we show that the noise term in GRAPH2SEQ's evolution rule (equation 1) is crucial for Theorem 1 to hold (proof in Appendix B.3).

**Proposition 1.** *Under any deterministic evolution rule of the form in equation 1, there exists a graph G which cannot be reconstructed exactly from its GRAPH2SEQ representation with arbitrarily high probability.*

The astute reader might observe that invertible, label-independent representations of graphs can be used to solve the graph isomorphism problem Schrijver (2003). However, Proposition 1 shows that GRAPH2SEQ cannot solve the graph isomorphism problem, as that would require a *deterministic* representation. Noise is necessary to *break symmetry* in the otherwise deterministic dynamical system. Observe that the time-series for a vertex in equation 1 depends *only* on the time-series of its neighboring nodes, not any explicit vertex identifiers. As a result, two graphs for which all vertices have exactly the same neighbors will have exactly the same representations, even though they may be structurally different. The proof of Proposition 1 illustrates this phenomenon for regular graphs.

### 3.2 FORMAL COMPUTATION MODEL

Although GRAPH2SEQ is an invertible representation of a graph, it is unclear how it compares to other GCNN representations in the literature. Below we define a formal computational model on graphs, called LOCAL-GATHER, that includes GRAPH2SEQ as well as a large class of GCNN representations in the literature. Abstracting different representations into a formal computational model allows us reason about the fundamental limits of these methods. We show that GCNNs with a fixed number of convolutional steps cannot compute certain functions over graphs, where a sequence-based representation such as GRAPH2SEQ is able to do so. For simplicity of notation, we consider undirected graphs in this section and in the rest of this paper.

**LOCAL-GATHER model**. Consider an undirected graph $G(V, E)$ on which we seek to compute a function $f : \mathcal{G} \to \mathbb{R}$, where $\mathcal{G}$ is the space of all undirected graphs. In the $k$-LOCAL-GATHER model, computations proceed in two rounds: In the *local step*, each vertex $v$ computes a representation $r(v)$ that depends only on the subgraph of vertices that are at a distance of at most $k$ from $v$. Following this, in the *gather step*, the function $f$ is computed by applying another function $g(\cdot)$ over the collection of vertex representations $\{r(v) : v \in V\}$. GRAPH2SEQ is an instance of the $\infty$-LOCAL-GATHER model, i.e., in GRAPH2SEQ each vertex representation $r(v)$ depends on the entire graph—not just on vertices a constant number of hops away from the vertex—regardless of the size of the graph. $\infty$ is used here to emphasize that the subgraph neighborhood on which each vertex representation depends is not constrained in size. For a specific graph $G$ with diameter $\Delta(G)$, it is also true that GRAPH2SEQ is $\Delta(G)$-local-gather. GCNNs that use localized filters with a global aggregation (e.g., Kipf & Welling (2016); Khalil et al. (2017)) also fit this model (proof in Appendix B.4).

**Proposition 2.** *The spectral GCNN representation in Kipf & Welling (2016) and the spatial GCNN representation in Khalil et al. (2017) belong to the $4$-LOCAL-GATHER model.*

**Fixed-length representations are insufficient**. We show below that for a fixed $k > 0$, no algorithm from the $k$-LOCAL-GATHER model can compute certain canonical graph functions exactly (proof in Appendix B.5).

**Theorem 2.** *For any fixed $k > 0$, there exists a function $f(\cdot)$ and an input graph instance $G$ such that no $k$-LOCAL-GATHER algorithm can compute $f(G)$ exactly.*

The graph used to prove Theorem 2 is an *unlabeled* graph, where nodes and edges do not have any attributes (e.g., node degree) included. Having labels on nodes or edges can reduce the local neighborhood size required to compute a function by revealing information about the graph. For example, with node degrees as labels, the local step of a $(k-1)$-LOCAL-GATHER algorithm can obtain information about the $k$-hop neighborhood of vertices (see Appendix B.5 for details). For the graph $G$ and function $f(\cdot)$ used in the proof of Theorem 2, we present a sequence-based representation (from the $\infty$-LOCAL-GATHER) in Appendix B.6 that is able to asymptotically compute $f(G)$. This example demonstrates that sequence-based representations are more powerful than fixed-length graph representations in the LOCAL-GATHER model. Further, it illustrates how a trained neural network can produce sequential representations that can be used to compute specific functions.

**GRAPH2SEQ and graph kernels.** Graph kernels (Yanardag & Vishwanathan, 2015; Vishwanathan et al., 2010; Kondor & Pan, 2016) are another popular method of representing graphs. The main idea here is to define or learn a vocabulary of substructures (e.g., graphlets, paths, subtrees), and use counts of these substructures in a graph as its representation. The Weisfeiler-Lehman (WL) graph kernel (Shervashidze et al., 2011; Weisfeiler & Lehman, 1968) is closest to GRAPH2SEQ. Starting with labels (e.g., vertex degree) on vertices, the WL kernel iteratively performs local label updates similar to equation 1 but typically using discrete functions/maps. The final representation consists of counts of these labels (i.e., a histogram) in the graph. Each label corresponds to a unique subtree pattern. However, the labels themselves are not part of any structured space, and cannot be used to compare the similarity of the subtrees they represent. Therefore, during testing if new unseen labels (or equivalently subtrees) are encountered the resulting representation may not generalize.

## 4 NEURAL NETWORK DESIGN

We consider a RL formulation for combinatorial optimization problems on graphs. RL is well-suited to such problems since the true 'labels' (i.e., the optimal solution) may be unavailable or hard to compute. Additionally, the objective functions in these problems can be used as natural reward signals. An RL approach has been explored in recent works Vinyals et al. (2015); Bello et al. (2016); Khalil et al. (2017) under different representation techniques. Our learning framework uses the GRAPH2SEQ representation. Fig. 1 shows our neural network architecture. We feed the trajectories output by GRAPH2SEQ for all the vertices into a recurrent neural network (specifically RNN-GRU), whose outputs are then aggregated by a feedforward network to select a vertex. Henceforth we call this the G2S-RNN neural network architecture. The key feature of this design is that the length of the sequential representation is *not* fixed; we vary it depending on the input instance. We show that our model is able to learn rules—for both generating the sequence and processing it with the RNN—that generalize to operate on long sequences. In turn, this translates to algorithmic solutions that scale to large graph sizes.

**Reinforcement learning model.** We consider a RL formulation in which vertices are chosen *one at a time*. Each time the RL agent chooses a vertex, it receives a reward. The goal of training is to learn a policy such that cumulative reward is maximized. We use $Q$-learning to train the network.

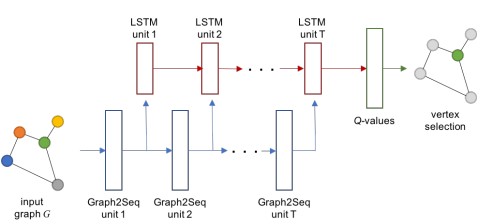

For input graph instance $G(V, E)$, a subset $S \subseteq V$ and $a \in V \setminus S$, this involves using a neural network to approximate a $Q$-*function* $Q(G, S, a)$. Here $S$ represents the set of vertices already picked. The neural network comprises of three modules: (1) *Graph2Seq*, that takes as input the graph $G$ and set $S$ of vertices chosen so far. It generates a sequence of vectors as output for each vertex. (2) *Seq2Vec* reads the sequences output of GRAPH2SEQ and summarizes it into one vector per vertex. (3) *Q-Network* takes the vector summary of each vertex $a \in V$ and outputs the esti-

Figure 1: The G2S-RNN neural network architecture. The GRU units process the representation output by GRAPH2SEQ, operating independently on each vertex's time-series.

mated $Q(G, S, a)$ value. The overall architecture is illustrated in Fig. 1. To make the network practical, we truncate the sequence outputs of Graph2Seq to a length of $T$. However the value of $T$ is not fixed, and is varied both during training and testing according to the size and complexity of the graph instances encountered; see § 5 for details. We describe each module below.

**Graph2Seq.** Let $\mathbf{x}_v(t)$ denote the state of vertex $v$ and $c_v(t)$ denote the binary variable that is one if $v \in S$ and zero otherwise, at time-step $t$ in the GRAPH2SEQ evolution. Then, the trajectory of each vertex $v \in V$ evolves as $\mathbf{x}_v(t + 1) = \text{relu}(\mathbf{W}_{G,1} \sum_{u \in \Gamma(v)} \mathbf{x}_u(t) + \mathbf{w}_{G,2} c_v(t) + \mathbf{w}_{G,3})$, for $t = 0, 1, \ldots, T - 1$. $\mathbf{W}_{G,1} \in \mathbb{R}^{d \times d}, \mathbf{w}_{G,2} \in \mathbb{R}^d, \mathbf{w}_{G,3} \in \mathbb{R}^d$ are trainable parameters, and $\mathbf{x}_v(0)$ is initialized to all-zeros for each $v \in V$.

**Seq2Vec and $Q$-Network.** The sequences $\{(\mathbf{x}_v(t))_{t=1}^T : v \in V\}$ are processed by GRU units (Chung et al., 2014) at the vertices. At time-step $t$, the recurrent unit at vertex $v$ computes as input a function that depends on (i) $\mathbf{x}_v(t)$, the embedding for node $v$, (ii) $\sum_{u \in \Gamma(v)} \mathbf{x}_u(t)$, the neighboring node

embeddings and (iii) $\sum_{u \in V} \mathbf{x}_u(t)$, a summary of embeddings of all nodes. This input is combined with the GRU's cell state $\mathbf{y}_v(t-1)$ to produce an updated cell state $\mathbf{y}_v(t)$. The cell state at the final time-step $\mathbf{y}_v(T)$ is the desired vector summary of the GRAPH2SEQ sequence, and is fed as input to the $Q$-network. We refer to Appendix C for equations on Seq2Vec. The $Q$-values are estimated as

$$\tilde{Q}(G, S, v) = \mathbf{w}_{Q,1}^T \mathrm{relu}(\mathbf{W}_{Q,2} \sum_{u \in V} \mathbf{y}_u(T)) + \mathbf{w}_{Q,3}^T \mathrm{relu}\left(\mathbf{W}_{Q,4} \mathbf{y}_v(T)\right), \tag{2}$$

with $\mathbf{W}_{Q,1}, \mathbf{W}_{Q,3} \in \mathbb{R}^{d \times d}$ and $\mathbf{w}_{Q,2}, \mathbf{w}_{Q,4} \in \mathbb{R}^d$ being learnable parameters. All transformation functions in the network leading up to equation 2 are differentiable. This makes the whole network differentiable, allowing us to train it end to end.

**Remark.** In Fig. 1 the GRAPH2SEQ RNN and the GRU RNN can also be thought of as two layers of a two-layer GRU. We have deliberately separated the two RNNs to highlight the fact that the sequence produced by GRAPH2SEQ (blue in Fig, 1) is the embedding of the graph, which is then read using a GRU layer (red in Fig. 1). Our architecture is not unique, and other designs for generating and/or reading the sequence are possible.

## 5 EVALUATIONS

In this section we present our evaluation results for GRAPH2SEQ. We address the following central questions: (1) How well does G2S-RNN scale? (2) How well does G2S-RNN generalize to new graph types? (3) Can we apply G2S-RNN to a variety of problems? (4) Does adversarial training improve scalability and generalization? To answer these questions, we experiment with G2S-RNN on three classical graph optimization problems: minimum vertex cover (MVC), max cut (MC) and maximum independent set (MIS). These are a set of problems well known to be NP-hard, and also greatly differ in their structure (Williamson & Shmoys, 2011). We explain the problems below.

**Minimum vertex cover.** The MVC of a graph $G(V, E)$ is the smallest cardinality set $S \subseteq V$ such that for every edge $(u, v) \in E$ at least one of $u$ or $v$ is in $S$. Approximation algorithms to within a factor 2 are known for MVC; however it cannot be approximated better than 1.3606 unless P=NP.
**Max cut.** In the MC problem, for an input graph instance $G(V, E)$ we seek a cut $(S, S^c)$ where $S \subseteq V$ such that the number of edges crossing the cut is maximized. This problem can be approximated within a factor 1.1383 of optimal, but not within 1.0684 unless P=NP.
**Maximum independent set.** For a graph $G(V, E)$ the MIS denotes a set $S \subseteq V$ of maximum cardinality such that for any $u, v \in S, (u, v) \notin E$. The maximum independent set is complementary to the minimum vertex cover—if $S$ is a MIC of $G$, then $V \setminus S$ is the MVC. However, from an approximation standpoint, MIS is hard to approximate within $n^{1-\epsilon}$ for any $\epsilon > 0$, despite constant factor approximation algorithms known for MVC.

**Heuristics compared.** In each problem, we compare G2S-RNN against: (1) Structure2Vec (Khalil et al., 2017), (2) GraphSAGE (Hamilton et al., 2017a) using (a) GCN, (b) mean and (c) pool aggregators, (3) WL kernel NN (Lei et al., 2017), (4) WL kernel embedding. In each of the above, the outputs of the last layer are fed to a $Q$-learning network as in §4. Unlike G2S-RNN the depth of the above neural network (NN) models are fixed across input instances. We also consider (5) Structure2Vec with the depth varied as in G2S-RNN (referred as S2V-dynamic henceforth; see Appendix D.1 for details), and the following well-known (non-learning based) heuristics for each problem: (6) *Greedy algorithms*, (7) *List heuristic*, (8) *Matching heuristic*. We refer to Appendix D.1 for details on these heuristics. We attempt to compute the optimal solution via the Gurobi optimization package (Gurobi Optimization, 2016). We run the Gurobi solver with a cut-off time of 240 s, and report performance in the form of approximation ratios relative to the solution found by the Gurobi solver. We do not compare against Deepwalk (Perozzi et al., 2014) or node2vec (Grover & Leskovec, 2016) since these methods are designed for obtaining vertex embeddings over a single graph. They are inappropriate for models that need to generalize over multiple graphs. This is because the vertex embeddings in these approaches can be arbitrarily rotated without a consistent 'alignment' across graphs. The number of parameters can also grow linearly with graph size. We refer to Hamilton et al. (2017a, Appendix D) for details.

**Training.** We train G2S-RNN, Structure2Vec, S2V-dynamic, GraphSAGE and WL kernel NN all on small Erdos-Renyi (ER) graphs of size 15, and edge probability 0.15. During training we truncate the G2S-RNN to 5 steps (i.e., $T = 5$, see Fig. 1). We refer to Appendix D.2 for more details on training.

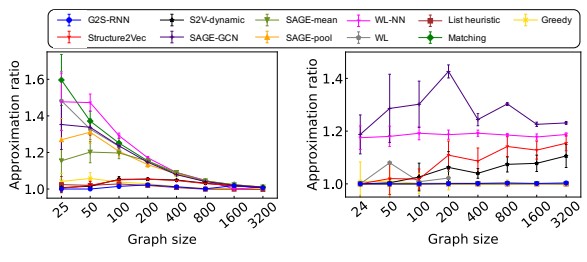

Figure 2: Scalability of G2S-RNN and GCNN in (a) Erdos-Renyi graphs (left), random bipartite graphs (right), and (b) on much larger graphs. The neural networks have been trained over the same graph types as the test graphs. Error bars show one standard deviation.

**Testing**. For any $T > 0$, let G2S-RNN($T$) denote the neural network (§4) in which GRAPH2SEQ is restricted to a sequence length of $T$. To test a graph $G(V, E)$, we feed $G$ as input to the networks G2S-RNN(1), G2S-RNN(2), ..., G2S-RNN($T_{\max}$), and choose the 'best' output as our final output. For each $T$, G2S-RNN($T$) outputs a solution set $S_T \subseteq V$. The best output corresponds to that $S_T$ having the maximum objective value. For e.g., in the case of MVC the $S_T$ having the smallest cardinality is the best output. This procedure is summarized in detail in Algorithm 2 in Appendix C. We choose $T_{\max} = 15$ in our experiments. The overall time-complexity of G2S-RNN is $O((|E| + |V|)T_{\max}^2|V|)$ as elaborated in Appendix C.3. To test generalization across graph types, we consider the following graph types: (1) ER graphs with edge probability 0.15; (2) random regular graphs with degree 4; (3) random bipartite graphs with equal sized partites and edge probability 0.75; (4) worst-case examples, such as the worst-case graph for the greedy heuristic on MVC, which has a $O(\log n)$ approximation factor Johnson (1973). (5) Two-dimensional grid graphs, in which the sides contain equal number of vertices. For each type, we test on graphs with number of vertices ranging from 25–3200 in exponential increments (except WL embedding which is restricted to size 100 or 200 since it is computationally expensive). Some of the graphs considered are dense—e.g., a 3200 node ER graph has 700,000 edges; a 3200 node random bipartite graph has 1.9 million edges. We also test on sparse ER and bipartite graphs of sizes 10,000 and 25000 with an average degree of 7.5.

## 5.1 SCALABILITY AND GENERALIZATION ACROSS GRAPH TYPES

**Scalability.** To test scalability, we train all the NN models on small graphs of a type, and test on larger graphs of the same type. For each NN model, we use the *same* trained parameters for all of the test examples in a graph type. We consider the MVC problem and train on: (1) size-15 ER graphs, and (2) size-20 random bipartite graphs. The models trained on ER graphs are then tested on ER graphs of sizes 25–3200; similarly the models trained on bipartite graphs are tested on bipartite graphs of sizes 24–3200. We present the results in Fig. 2(a). We have also included non-learning-based heuristics for reference. In both ER and bipartite graphs, we observe that G2S-RNN generalizes well to graphs of size roughly 25 through 3200, even though it was trained on size-15 graphs. The fixed-depth NN models, however, either generalize well on only one of the two types (e.g., Structure2Vec performs well on ER graphs, but not on bipartite graphs) or do not generalize in both types. S2V-dynamic exhibits performance considerably better than Structure2Vec. This is because S2V-dynamic is also using the entire sequence of Structure2Vec embeddings for each vertex (albeit implicitly). Therefore, this method is a different instantiation of our idea of using sequences as node embeddings. In particular, like G2S-RNN, this method also considers neighborhoods of different sizes around each vertex for different graphs. The only difference is that G2S-RNN additionally uses a GRU to process the sequence. G2S-RNN generalizes well to even larger graphs. Fig. 2(b) presents results of testing on size 10,000 and 25,000 ER and random bipartite graphs. We observe the vertex cover output by G2S-RNN is at least 100 nodes fewer than Structure2Vec.

**Generalization across graph types.** Next we test how the models generalize to different graph types. We train the models on size-15 ER graphs, and test them on three graph types: (i) worst-case graphs, (ii) random regular graphs, and (iii) random bipartite graphs. For each graph type, we vary the graph size from 25 to 3200 as before. Fig. 3 plots results for the different baselines. In general, G2S-RNN has a performance that is within 10% of the optimal, across the range of graph types and sizes considered. The other NN baselines demonstrate behavior that is not consistent and have certain classes of graph types/sizes where they perform poorly.

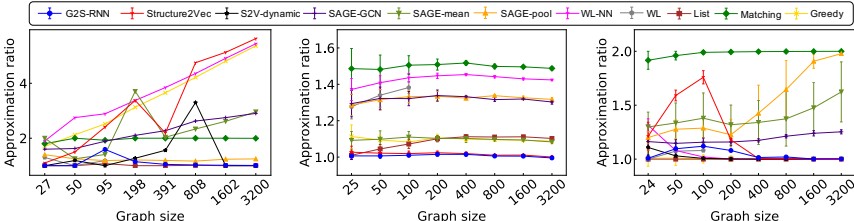

Figure 3: Minimum vertex cover in worst-case graphs for greedy (left), random regular graphs (center), and random bipartite graphs (right) using models trained on size-15 Erdos-Renyi graphs. Each point shows that average and standard deviation for 20 randomly sampled graphs of that type. The worst-case greedy graph type, however, has only one graph at each size; hence, the results show outliers for this graph type.

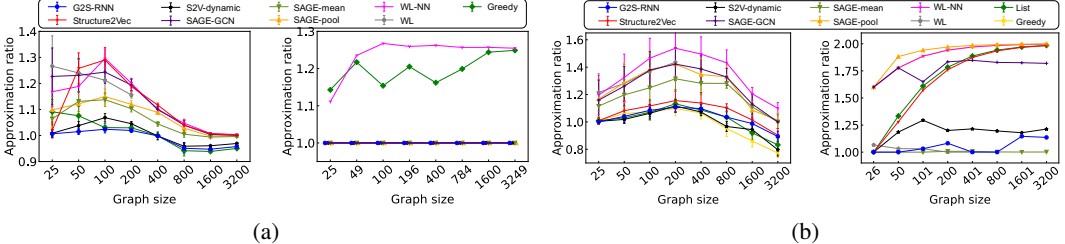

Figure 4: (a) Max cut in Erdos-Renyi graphs (left), Grid graphs (right). (b) Maximum independent set in Erdos-Renyi graphs (left), structured bipartite graphs (right). Some approximation ratios are less than 1 due to the time cut-off of the Gurobi solver.

**Adversarial training.** We also trained G2S-RNN on a certain class of adversarial 'hard' examples for minimum vertex cover, and observed further improvements in generalization. We refer to Appendix D for details and results of this method.

## 5.2 OTHER PROBLEMS: MC AND MIS

We test and compare G2S-RNN on the MC and MIS problems. As in MVC, our results demonstrate consistently good scalability and generalization of G2S-RNN across graph types and sizes. As before, we train the NN models on size-15 ER graphs ($p = 0.15$) and test on different graphs.

**Max cut.** We test on (1) ER graphs, and (2) two-dimensional grid graphs. For each graph type, we vary the number of vertices in the range 25–3200, and use the same trained model for all of the tests. The results of our tests are presented in Fig. 4(a). We notice that for both graph types G2S-RNN is able to achieve an approximation less that $1.04$ times the (timed) integer program output.

**Maximum independent set.** We test on (1) ER graphs, and (2) worst-case bipartite graphs for the greedy heuristic. The number of vertices is varied in the range 25–3200 for each graph type. We present our results in Fig. 4(b). In ER graphs, G2S-RNN shows a reasonable consistency in which it is always less than 1.10 times the (timed) integer program solution. In the bipartite graph case we see a performance within 8% of optimal across all sizes.

## 6 CONCLUSION

We proposed GRAPH2SEQ that represents vertices of graphs as infinite time-series of vectors. The representation melds naturally with modern RNN architectures that take time-series as inputs. We applied this combination to three canonical combinatorial optimization problems on graphs, ranging across the complexity-theoretic hardness spectrum. Our empirical results best state-of-the-art approximation algorithms for these problems on a variety of graph sizes and types. In particular, GRAPH2SEQ exhibits significantly better scalability and generalization than existing GCNN representations in the literature. An open direction involves a more systematic study of the capabilities of GRAPH2SEQ across the panoply of graph combinatorial optimization problems, as well as its performance in concrete (and myriad) downstream applications. Another open direction involves interpreting the policies learned by GRAPH2SEQ to solve specific combinatorial optimization problems (e.g., as in LIME (Ribeiro et al., 2016)). A detailed analysis of the GRAPH2SEQ dynamical system to study the effects of sequence length on the representation is also an important direction.

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

## A    BACKGROUND: GRAPH CONVOLUTIONAL NEURAL NETWORKS

An ideal graph representation is one that captures all innate structures of the graph relevant to the task at hand, and moreover can also be learned via gradient descent methods. However, this is challenging since the relevant structures could range anywhere from local attributes (example: node degrees) to long-range dependencies spanning across a large portion of the graph (example: does there exist a path between two vertices) (Kuhn et al., 2016). Such broad scale variation is also a well-known issue in computer vision (image classification, segmentation etc.), wherein convolutional neural network (CNN) designs have been used quite successfully (Krizhevsky et al., 2012). Perhaps motivated by this success, recent research has focused on generalizing the traditional CNN architecture to develop designs for graph convolutional neural networks (GCNN) (Bruna et al., 2014; Niepert et al., 2016). By likening the relationship between adjacent pixels of an image to that of adjacent nodes in a graph, the GCNN seeks to emulate CNNs by defining localized 'filters' with shared parameters.

Current GCNN filter designs can be classified into one of two categories: *spatial* (Kipf & Welling, 2016; Khalil et al., 2017; Nowak et al., 2017), and *spectral* (Defferrard et al., 2016). For an integral hyper-parameter $K \geq 0$, filters in either category process information from a $K$-local neighborhood surrounding a node to compute the output. Here we consider localized spectral filters such as proposed in Defferrard et al. (2016). The difference between the spatial and spectral versions arises in the precise way in which the aggregated local information is combined.

*Spatial GCNN.* For input feature vector $\mathbf{x}_v$ at each node $v \in V$ of a graph $G$, a spatial filtering operation is the following:

$$\mathbf{y}_v = \sigma \left( \sum_{k=0}^{K-1} \mathbf{W}_k \left( \sum_{u \in V} (\tilde{\mathbf{A}}^k)_{v,u} \mathbf{x}_u \right) + \mathbf{b}_0 \right) \quad \forall v \in V, \tag{3}$$

where $\mathbf{y}_v$ is the filter output, $\mathbf{W}_k, k = 1, \ldots, K$ and $\mathbf{b}_0$ are learnable parameters, and $\sigma$ is a non-linear activation function that is applied element-wise. $\tilde{\mathbf{A}} = \mathbf{D}^{-1/2}\mathbf{A}\mathbf{D}^{-1/2}$ is the normalized adjacency matrix, and $\mathbf{D}$ is the diagonal matrix of vertex degrees. Use of un-normalized adjacency matrix is also common. The $k$-power of the adjacency matrix selects nodes a distance of at most $k$ hops from $v$. ReLU is a common choice for $\sigma$. We highlight two aspects of spatial GCNNs: (i) the feature vectors are aggregated from neighboring nodes directly specified through the graph topology, and (ii) the aggregated features are summarized via an addition operation.

*Spectral GCNN.* Spectral GCNNs use the notion of graph Fourier transforms to define convolution operation as the inverse transform of multiplicative filtering in the Fourier domain. Since this is a non-local operation potentially involving data across the entire graph, and moreover it is computationally expensive to compute the transforms, recent work has focused on approximations to produce a local spectral filter of the form

$$\mathbf{y}_v = \sigma \left( \sum_{k=0}^{K-1} \mathbf{W}'_k \left( \sum_{u \in V} (\tilde{\mathbf{L}}^k)_{v,u} \mathbf{x}_u \right) + \mathbf{b}'_0 \right) \quad \forall v \in V, \tag{4}$$

where $\tilde{\mathbf{L}} = \mathbf{I} - \tilde{\mathbf{A}}$ is the normalized Laplacian of the graph, $(\tilde{\mathbf{L}}^k)_{v,u}$ denotes the entry at the row corresponding to vertex $v$ and column corresponding to vertex $u$ in $\tilde{\mathbf{L}}^k$, and $\mathbf{W}'_k, \mathbf{b}'_0$ are parameters (Defferrard et al., 2016; Kipf & Welling, 2016). As in the spatial case, definitions using unnormalized version of Laplacian matrix are also used. $\sigma$ is typically the identity function here. The function in equation 4 is a local operation because the $k$-th power of the Laplacian, at any row $v$, has a support no larger than the $k$-hop neighborhood of $v$. Thus, while the aggregation is still localized, the feature vectors are now weighted by the entries of the Laplacian before summation.

*Spectral and Spatial GCNN are equivalent.* The distinction between spatial and spectral convolution designs is typically made owing to their seemingly different definitions. However we show that both designs are mathematically equivalent in terms of their representation capabilities.

**Proposition 3.** *Consider spatial and spectral filters in equation 3 and equation 4, using the same nonlinear activation function $\sigma$ and $K$. Then, for graph $G(V, E)$, for any choice of parameters $\mathbf{W}_k$ and $\mathbf{b}_0$ for $k = 1, \ldots, K$ there exists parameters $\mathbf{W}'_k$ and $\mathbf{b}'_0$ such that the filters represent the same transformation function, and vice-versa.*

*Proof.* Consider a vertex set $V = \{1, 2, \ldots, n\}$ and $d$-dimensional vertex states $\mathbf{x}_i$ and $\mathbf{y}_i$ at vertex $i \in V$. Let $\mathbf{X} = [\mathbf{x}_1, \ldots, \mathbf{x}_n]$ and $\mathbf{Y} = [\mathbf{y}_1, \ldots, \mathbf{y}_n]$ be the matrices obtained by concatenating the state vectors of all vertices. Then the spatial transformation function of equation 3 can be written as

$$\mathbf{Y} = \sigma \left( \sum_{k=0}^{K-1} \mathbf{W}_k \mathbf{X} \tilde{\mathbf{A}}^k + \mathbf{b}_0 \mathbf{1}^T \right), \tag{5}$$

while the spectral transformation function of equation 4 can be written as

$$\mathbf{Y} = \sigma \left( \sum_{k=0}^{K-1} \mathbf{W}'_k \mathbf{X} \tilde{\mathbf{L}}^k + \mathbf{b}'_0 \mathbf{1}^T \right) \tag{6}$$

$$= \sigma \left( \sum_{k=0}^{K-1} \mathbf{W}'_k \mathbf{X} (\mathbf{I} - \tilde{\mathbf{A}})^k + \mathbf{b}'_0 \mathbf{1}^T \right) \tag{7}$$

$$= \sigma \left( \sum_{k=0}^{K-1} \mathbf{W}'_k \mathbf{X} \sum_{i=0}^{k} \binom{k}{i} (-1)^{k-i} \tilde{\mathbf{A}}^i + \mathbf{b}'_0 \mathbf{1}^T \right) \tag{8}$$

$$= \sigma \left( \sum_{k=0}^{K-1} \left( \sum_{i=k}^{K-1} \mathbf{W}'_i \binom{i}{k} (-1)^{i-k} \right) \mathbf{X} \tilde{\mathbf{A}}^k + \mathbf{b}'_0 \mathbf{1}^T \right). \tag{9}$$

The equation 7 follows by the definition of the normalized Laplacian matrix, and equation 8 derives from binomial expansion. To make the transformation in equation 5 and equation 9 equal, we can set

$$\sum_{i=k}^{K-1} \mathbf{W}'_i \binom{i}{k} (-1)^{i-k} = \mathbf{W}_k, \quad \forall\, 0 \leq k \leq K-1, \tag{10}$$

and check if there are any feasible solutions for the primed quantities. Clearly there are, with one possible solution being $\mathbf{b}'_0 = \mathbf{b}_0$ and

$$\mathbf{W}'_{K-1} = \mathbf{W}_{K-1} \tag{11}$$

$$\mathbf{W}'_k = \mathbf{W}_k - \sum_{i=k+1}^{K-1} \mathbf{W}'_i \binom{i}{k} (-1)^{i-k}, \tag{12}$$

$\forall\, 0 \leq k \leq K-2$. Thus for any choice of values for $\mathbf{W}_k, \mathbf{b}_0$ for $k = 0, \ldots, K-1$ there exists $\mathbf{W}'_k, \mathbf{b}'_0$ for $k = 0, \ldots, K-1$ such that the spatial and spectral transformation functions are equivalent. The other direction (when $\mathbf{W}'_k$ and $\mathbf{b}_0$ are fixed), is similar and straightforward. □

Depending on the application, the convolutional layers may be supplemented with pooling and coarsening layers that summarize outputs of nearby convolutional filters to form a progressively more compact spatial representation of the graph. This is useful in classification tasks where the desired output is one out of a few possible classes (Bruna et al., 2014). For applications requiring decisions at a per-node level (e.g. community detection), a popular strategy is to have multiple repeated convolutional layers that compute vector representations for each node, which are then processed to make a decision (Khalil et al., 2017; Bruna & Li, 2017; Nowak et al., 2017). The conventional wisdom here is to have as many layers as the diameter of the graph, since filters at each layer aggregate information only from nearby nodes. Such a strategy is sometimes compared to the message passing algorithm (Gilmer et al., 2017), though the formal connections are not clear as noted in Nowak et al. (2017). Finally the GCNNs described so far are all end-to-end differentiable and can be trained using mainstream techniques for supervised, semi-supervised or reinforcement learning applications.

Other lines of work use ideas inspired from word embeddings for graph representation (Grover & Leskovec, 2016; Perozzi et al., 2014). Post-GCNN representation, LSTM-RNNs have been used to analyze time-series data structured over a graph. Seo et al. (2016) propose a model which combines GCNN and RNN to predict moving MNIST data. Liang et al. (2016) design a graph LSTM for semantic object parsing in images.

# B    DETAILS ON SECTION 3 AND PROOFS

## B.1    INCLUDING VERTEX AND EDGE FEATURES IN GRAPH2SEQ

GRAPH2SEQ naturally allows for vertex and edge features to be included as part of the evolution rule in equation 1, by adding extra terms within the ReLU function. If $\mathbf{y}_e$ denotes the real-valued features of edge $e$, and $\mathbf{z}_v$ denotes the features of vertex $v$, then the evolution rule can be modified as

$$\mathbf{x}_v(t+1) = \text{relu}(\mathbf{W}_1 \sum_{u \in \Gamma(v)} \mathbf{x}_u(t) + \mathbf{W}_2 \sum_{u \in \eta(v)} \mathbf{y}_e + \mathbf{W}_3 \mathbf{z}_v + \mathbf{b}_1) + \mathbf{n}_v(t+1), \quad (13)$$

where $\eta(v)$ denotes the edges incident to vertex $v$, and $\mathbf{W}_1, \mathbf{W}_2, \mathbf{W}_3, \mathbf{b}_1$ are parameters. Another way to include edge features is to transform the graph with edge features into a new (larger) graph where there are only vertex features and no edge features. This is done by converting the original graph into a new bipartite graph where one partite corresponds to vertices of the original graph, and the other partite corresponds to edges of the original graph. Each edge-node in the bipartite graph is connected to the two vertex-nodes that constitute its end points in the original graph. The edge-nodes have the edge features of the original graph, while the vertex-nodes have the vertex features.

## B.2    PROOF OF THEOREM 1

*Proof.* Consider a GRAPH2SEQ trajectory on graph $G(V, E)$ according to equation 1 in which the vertex states are initialized randomly from some distribution. Let $\mathbf{X}_v(t)$ (resp. $\mathbf{x}_v(t)$) denote the random variable (resp. realization) corresponding to the state of vertex $v$ at time $t$. For time $T > 0$ and a set $S \subseteq V$, let $\mathbf{X}_S^T$ denote the collection of random variables $\{\mathbf{X}_v(t) : v \in S, 0 \le t \le T\}$; $\mathbf{x}_V^T$ will denote the realizations.

An information theoretic estimator to output the graph structure by looking at the trajectory $\mathbf{X}_V^T$ is the directed information graph considered in Quinn et al. (2015). Roughly speaking, the estimator evaluates the conditional directed information for every pair of vertices $u, v \in V$, and declares an edge only if it is positive (see Definition 3.4 in Quinn et al. (2015) for details). Estimating conditional directed information efficiently from samples is itself an active area of research Quinn et al. (2011), but simple plug-in estimators with a standard kernel density estimator will be consistent. Since the theorem statement did not specify sample efficiency (i.e., how far down the trajectory do we have to go before estimating the graph with a required probability), the inference algorithm is simple and polynomial in the length of the trajectory. The key question is whether the directed information graph is indeed the same as the underlying graph $G$. Under some conditions on the graph dynamics (discussed below in Properties 1–3), this holds and it suffices for us to show that the dynamics generated according to equation 1 satisfies those conditions.

**Property 1.** *For any $T > 0$, $P_{\mathbf{X}_V^T}(\mathbf{x}_V^T) > 0$ for all $\mathbf{x}_V^T$.*

This is a technical condition that is required to avoid degeneracies that may arise in deterministic systems. Clearly GRAPH2SEQ's dynamics satisfies this property due to the additive i.i.d. noise in the transformation functions.

**Property 2.** *The dynamics is strictly causal, that is $P_{\mathbf{X}_V^T}(\mathbf{x}_V^T)$ factorizes as $\prod_{t=0}^{T} \prod_{v \in V} P_{\mathbf{X}_v(t)|\mathbf{X}_V^{t-1}}(\mathbf{x}_v(t)|\mathbf{x}_V^{t-1})$.*

This is another technical condition that is readily seen to be true for GRAPH2SEQ. The proof also follows from Lemma 3.1 in Quinn et al. (2015).

**Property 3.** *$G$ is the minimal generative model graph for the random processes $\mathbf{X}_v(t), v \in V$.*

Notice that the transformation operation equation 1 in our graph causes $\mathbf{X}_V^T$ to factorize as

$$P_{\mathbf{X}_V^T}(\mathbf{x}_V^T) = \prod_{t=0}^{T} \prod_{v \in V} P_{\mathbf{X}_v(t)|\mathbf{X}_{\Gamma(v)}^{t-1}}(\mathbf{x}_v(t)|\mathbf{x}_{\Gamma(v)}^{t-1}) \quad (14)$$

for any $T > 0$, where $\Gamma(v)$ is the set of neighboring vertices of $v$ in $G$. Now consider any other graph $G'(V, E')$. $G'$ will be called a minimal generative model for the random processes $\{\mathbf{X}_v(t) : v \in V, t \geq 0\}$ if

(1) there exists an alternative factorization of $P_{\mathbf{X}_V^T}(\mathbf{x}_V^T)$ as

$$P_{\mathbf{X}_V^T}(\mathbf{x}_V^T) = \prod_{t=0}^{T} \prod_{v \in V} P_{\mathbf{X}_v(t)|\mathbf{X}_{\Gamma'(v)}^{t-1}}(\mathbf{x}_v(t)|\mathbf{x}_{\Gamma'(v)}^{t-1}) \tag{15}$$

for any $T > 0$, where $\Gamma'(v)$ is the set of neighbors of $v$ in $G'$, and

(2) there does not exist any other graph $G''(V, E'')$ with $E'' \subset E$ and a factorization of $P_{\mathbf{X}_V^T}(\mathbf{x}_V^T)$ as $\prod_{t=0}^{T} \prod_{v \in V} P_{\mathbf{X}_v(t)|\mathbf{X}_{\Gamma''(v)}^{t-1}}(\mathbf{x}_v(t)|\mathbf{x}_{\Gamma''(v)}^{t-1})$ for any $T > 0$, where $\Gamma''(v)$ is the set of neighbors of $v$ in $G''$.

Intuitively, a minimal generative model is the smallest spanning graph that can generate the observed dynamics. To show that $G(V, E)$ is indeed a minimal generative model, let us suppose the contrary and assume there exists another graph $G'(V, E')$ with $E' \subset E$ and a factorization of $P_{\mathbf{X}_V^T}(\mathbf{x}_V^T)$ as in equation 15. In particular, let $v$ be any node such that $\Gamma'(v) \subset \Gamma(v)$. Then by marginalizing the right hand sides of equation 14 and equation 15, we get

$$P_{\mathbf{X}_v(1)|\mathbf{X}_{\Gamma(v)}^0}(\mathbf{x}_v(1)|\mathbf{x}_{\Gamma(v)}^0) = P_{\mathbf{X}_v(1)|\mathbf{X}_{\Gamma'(v)}^0}(\mathbf{x}_v(1)|\mathbf{x}_{\Gamma'(v)}^0). \tag{16}$$

Note that equation 16 needs to hold for all possible realizations of the random variables $\mathbf{X}_v(1), \mathbf{X}_{\Gamma(v)}^0$ and $\mathbf{X}_{\Gamma'(0)}^0$. However if the parameters $\mathbf{\Theta}_0$ and $\Theta_1$ in equation 1 are generic, this is clearly not true. To see this, let $u \in \Gamma(v) \backslash \Gamma'(v)$ be any vertex. By fixing the values of $\mathbf{x}_v(1), \mathbf{x}_{\Gamma(v)\backslash\{u\}}^0$ it is possible to find two values for $\mathbf{x}_u(0)$, say $\mathbf{a_1}$ and $\mathbf{a}_2$, such that

$$\text{ReLU}\left(\mathbf{\Theta}_0\left(\sum_{i \in \Gamma(v)\backslash\{u\}} \mathbf{x}_i(0) + \mathbf{a}_1\right) + \Theta_1\right)$$

$$\neq \text{ReLU}\left(\mathbf{\Theta}_0\left(\sum_{i \in \Gamma(v)\backslash\{u\}} \mathbf{x}_i(0) + \mathbf{a}_2\right) + \Theta_1\right). \tag{17}$$

As such the Gaussian distributions in these two cases will have different means. However the right hand side Equation equation 16 does not depend on $\mathbf{x}_u(0)$ at all, resulting in a contradiction. Thus $G$ is a minimal generating function of $\{\mathbf{X}_v(t) : v \in V, t \geq 0\}$. Thus Property 3 holds as well. Now the result follows from the following Theorem.

**Theorem 3** (Theorem 3.6, Quinn et al. (2015)). *If Properties 1, 2 and 3 are satisfied, then the directed information graph is equivalent to the graph $G$.*

$\square$

### B.3 PROOF OF PROPOSITION 1

*Proof.* Consider 4-regular graphs $R_1$ and $R_2$ with vertices $\{0, 1, \ldots, 7\}$ and edges $\{(0, 3), (0, 5), (0, 6), (0, 7), (1, 2), (1, 4), (1, 6), (1, 7), (2, 3), (2, 5), (2, 6), (3, 4), (3, 5), (4, 5), (4, 7), (6, 7)\}$ and $\{(0, 1), (0, 2), (0, 4), (0, 7), (1, 4), (1, 5), (1, 6), (2, 3), (2, 4), (2, 7), (3, 5), (3, 6), (3, 7), (4, 6), (5, 6), (5, 7)\}$ respectively. Then under a deterministic evolution rule, since $R_1$ and $R_2$ are 4-regular graphs, the trajectory will be identical at all nodes across the two graphs. However the graphs $R_1$ and $R_2$ are structurally different. For e.g., $R_1$ has a minimum vertex cover size of 5, while for $R_2$ it is 6. As such, if any one of the graphs ($R_1$, say) is provided as input to be represented, then from the representation it is impossible to exaclty recover $R_1$'s structure. $\square$

### B.4 PROOF OF PROPOSITION 2

*Proof.* Kipf & Welling (2016) use a two layer graph convolutional network, in which each layer uses convolutional filters that aggregate information from the immediate neighborhood of the vertices.

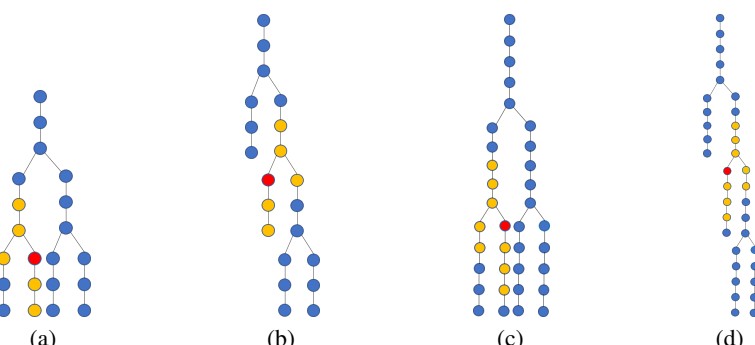

Figure 5: Example to illustrate $k$-LOCAL-GATHER algorithms are insufficient for computing certain functions. Corresponding vertices in the two trees in (a) and (b) above have similar $k = 2$ local neighborhoods, but the trees have minimum vertex cover of different sizes (Figure 6). Similarly the trees in (c) and (d) contain nodes with similar $k = 3$ local neighborhoods, but have differing minimum vertex cover sizes (Figure 6).

This corresponds to a 2-local representation function $r(\cdot)$ in our computational model. Following this step, the values at the vertices are aggregated using softmax to compute a probability score at each vertex. Since this procedure is independent of the structure of the input graph, it is a valid gathering function $g(\cdot)$ in LOCAL-GATHER and the overall architecture belongs to a 2-LOCAL-GATHER model.

Similarly, Khalil et al. (2017) also consider convolutional layers in which the neurons have a spatial locality of one. Four such convolutional layers are cascaded together, the outputs of which are then processed by a separate $Q$-learning network. Such a neural architecture is an instance of the 4-LOCAL-GATHER model. □

### B.5 Proof of Theorem 2

*Proof.* Consider a family $\mathcal{G}$ of undirected, unweighted graphs. Let $f : \mathcal{G} \to \mathbb{Z}$ denote a function that computes the size of the minimum vertex cover of graphs from $\mathcal{G}$. For $k > 0$ fixed, let ALG denote any algorithm from the $k$-LOCAL-GATHER model, with a representation function $r_{\texttt{ALG}}(\cdot)$ and aggregating function $g_{\texttt{ALG}}(\cdot)$.[1] We present two graphs $G_1$ and $G_2$ such that $f(G_1) \neq f(G_2)$, but the set of computed states $\{r_{\texttt{ALG}}(v) : v \in G_i\}$ is the same for both the graphs $(i = 1, 2)$. Now, since the gather function $g_{\texttt{ALG}}(\cdot)$ operates only on the set of computed states (by definition of our model), this implies ALG cannot distinguish between $f(G_1)$ and $f(G_2)$, thus proving our claim.

For simplicity, we first describe the proof for $k = 2$. We consider the graphs $G_1$ and $G_2$ as shown in Fig. 5(a) and 5(b) respectively. To construct these graphs, we first consider binary trees $B_1$ and $B_2$ each having 7 nodes. $B_1$ is a completely balanced binary tree with a depth of 2, whereas $B_2$ is a completely imbalanced binary tree with a depth of 3. Now, to get $G_1$ and $G_2$, we replace each node in $B_1$ and $B_2$ by a chain of 3 nodes. At each location in $B_i$ $(i = 1, 2)$, the head of the chain of nodes connects to the tail of the parent's chain of nodes. Fig. 5(a) and 5(b) illustrate the trees obtained by this process for $k = 2$.

The sizes of the minimum vertex cover of $G_1$ and $G_2$ are 9 and 10 respectively (Fig. 6). However, there exists a one-to-one mapping between the vertices of $G_1$ and the vertices of $G_2$ such that the 2-hop neighborhood around corresponding vertices in $G_1$ and $G_2$ are the same. For example, in Fig. 5(a) and 5(b) the pair of nodes shaded in red have an identical 2-hop neighborhood (shaded in yellow). As such, the representation function $r_{\texttt{ALG}}(\cdot)$ – which for any node depends only on its $k$-hop neighborhood – will be the same for corresponding pairs of nodes in $G_1$ and $G_2$.

Finally, the precise mapping between pairs of nodes in $G_1$ and $G_2$ is obtained as follows. First consider a simple mapping between pairs of nodes in $B_1$ and $B_2$ in which (i) the 4 leaf nodes in $B_1$ are mapped to the leaf nodes in $B_2$, (ii) the root of $B_1$ is mapped to the root of $B_2$ and (iii) the

---

[1]See beginning of Section 3 for explanations of $r(\cdot)$ and $g(\cdot)$.

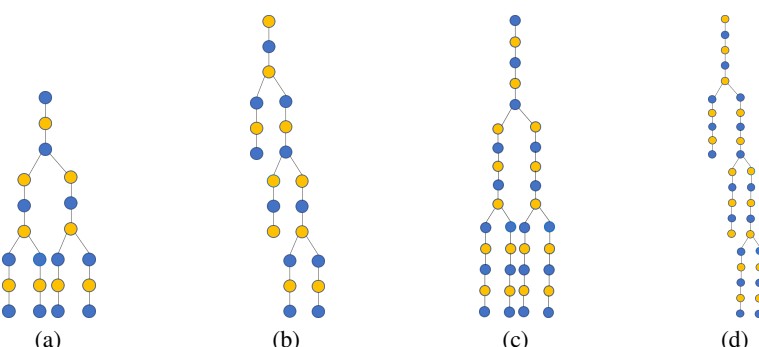

Figure 6: Minimum vertex cover solution for the trees shown in Fig. 5. The vertices that belong to the vertex cover are shaded in yellow.

2 interior nodes of $B_1$ are mapped to the interior nodes of $B_2$. We generalize this mapping to $G_1$ and $G_2$ in two steps: (1) mapping chains of nodes in $G_1$ to chains of nodes in $G_2$, according to the $B_1 - B_2$ map, and (2) within corresponding chains of nodes, we map nodes according to order (head-to-head, tail-to-tail, etc.).

For a general $k$, we replace each node in $B_1$ and $B_2$ by a chain of $l$ nodes, where $l$ is any odd number greater than $k$. It can be shown that the trees constructed this way have minimum vertex cover sizes that differ by 1. Fig. 5(c) and 5(d) illustrate the trees obtained for $k = 3$, where the chains contain 5 nodes each. The tree in Fig. 5(c) has a minimum vertex cover size of 16 while it is 17 for the tree in Fig. 5(d). The mapping between the vertices in the two trees can be done analogously as described in the $k = 2$ case above. □

**Remark on node labels.** On labeled graphs, where attributes such as node degree are included as part of each node, the local neighborhood size required to compute a function can be much smaller. For example, if nodes are labeled by their degrees, the local step in a $k$-LOCAL-GATHER algorithm can cause node representations to actually depend on the $(k+1)$ hop neighborhood around each node. This is because the node degree labels already reveal information about the 1-hop neighborhood of each node. As a result, a $(k-1)$-LOCAL-GATHER algorithm may suffice on such a labeled graph, where a $k$-LOCAL-GATHER algorithm was originally required. In the extreme case, node labels can encode the entire structure of the graph (e.g., if adjacency matrix is used as a node label). In this case, even a 0-LOCAL-GATHER algorithm will be able to exactly compute any function over the graph because the entire graph can be inferred just by looking at a node's label. The graphs we have considered in the proof of Theorem 2 are unlabeled graphs. In unlabeled graphs, attributes such as node degree are not inherently part of a node and need to be explicitly computed.

### B.6 SEQUENTIAL HEURISTIC TO COMPUTE MVC ON TREES

Consider any unweighted, undirected tree $T$. We let the state at any node $v \in T$ be represented by a two-dimensional vector $[x_v, y_v]$. For any $v \in T$, $x_v$ takes values over the set $\{-\epsilon, +1\}$ while $y_v$ is in $\{-1, 0, \epsilon\}$. Here $\epsilon$ is a parameter that we choose to be less than one over the maximum degree of the graph. Semantically $x_v$ stands for whether vertex $v$ is 'active' ($x_v = +1$) or 'inactive' ($x_v = -\epsilon$). Similarly $y_v$ stands for whether $v$ has been selected to be part of the vertex cover ($y_v = +\epsilon$), has not been selected to be part of the cover ($y_v = -1$), or a decision has not yet been made ($y_v = 0$). Initially $x_v = -\epsilon$ and $y_v = 0$ for all vertices. The heuristic proceeds in rounds, wherein at each round any vertex $v$ updates its state $[x_v, y_v]$ based on the state of its neighbors as shown in Algorithm 1.

The update rules at vertex $v$ are (1) if $v$ is a leaf or if at least one of $v$'s neighbors are active, then $v$ becomes active; (2) if $v$ is active, and if at least one of $v$'s active neighbors have not been chosen in the cover, then $v$ is chosen to be in the cover; (3) if all of $v$'s neighbors are inactive, then $v$ remains inactive and no decision is made on $y_v$.

---

**Algorithm 1** Sequential heuristic to compute minimum vertex cover on a tree.

---

**Input:** Undirected, unweighted tree $T$; Number of rounds `NumRounds`
**Output:** Size of minimum vertex cover on $T$
$x_v(0) \leftarrow -\epsilon$ for all $v \in T$ {$x_v(i)$ is $x_v$ at round $i$}
$y_v(0) \leftarrow 0$ for all $v \in T$ {$y_v(i)$ is $y_v$ at round $i$}
// Computing the representation $r(v)$ for each $v \in T$
**for** i from 1 **to** `NumRounds` **do**
  At each vertex $v$:
  **if** $\sum_{u \in \Gamma(v)} x_u(i-1) \geq -\epsilon$ **then**
    $x_v(i) \leftarrow +1$
    **if** $\sum_{u \in \Gamma(v)} y_u(i-1) < 0$ **then**
      $y_v(i) \leftarrow \epsilon$
    **else**
      $y_v(i) \leftarrow -1$
    **end if**
  **else**
    $x_v(i) \leftarrow -\epsilon$ and $y_v(i) \leftarrow 0$
  **end if**
**end for**
// Computing the aggregating function $g(\{r(v) : v \in T\})$
$\bar{y}_v \leftarrow \left( \sum_{i=1}^{\texttt{NumRounds}} (y_v(i) + 1)/(1 + \epsilon) \right) / \texttt{NumRounds}$
**return** $\sum_{v \in T} \bar{y}_v$

---

At the end of the local computation rounds, the final vertex cover size is computed by first averaging the $y_v$ time-series at each $v \in T$ (with translation, and scaling as shown in Algorithm 1), and then summing over all vertices.

---

**Algorithm 2** Testing procedure of GRAPH2SEQ on a graph instance.

---

**Input:** graph $G$, trained parameters, objective $f : G \rightarrow \mathbb{R}$ we seek to maximize, maximum sequence length $T_{\max}$
**Output:** solution set $S \subseteq V$
$S_{\text{opt}} \leftarrow \{\}, v_{\text{opt}} \leftarrow 0$ {initialize}
**for** $T$ from 1 **to** $T_{\max}$ **do**
   $S \leftarrow$ solution returned by GRAPH2SEQ $(T)$
   **if** $f(S) > v_{\text{opt}}$ **then**
      $S_{\text{opt}} \leftarrow S$
      $v_{\text{opt}} \leftarrow f(S)$
   **end if**
**end for**
**return** $S_{\text{opt}}$

---

## C  TRAINING AND TESTING ALGORITHMS

### C.1  REINFORCEMENT LEARNING FORMULATION

Let $G(V, E)$ be an input graph instance for the optimization problems mentioned above. Note that the solution to each of these problems can be represented by a set $S \subseteq V$. In the case of the minimum vertex cover (MVC) and maximum independent set (MIS), the set denotes the desired optimal cover and independent set respectively; for max cut (MC) we let $(S, S^c)$ denote the optimal cut. For the following let $f : 2^V \rightarrow \mathbb{R}$ be the objective function of the problem (i.e., MVC, MC or MIS) that we want to maximize, and let $\mathcal{F} \subseteq 2^V$ be the set of feasible solutions.

**Dynamic programming formulation.** Now, consider a dynamic programming heuristic in which the subproblems are defined by the pair $(G, S)$, where $G$ is the graph and $S \subseteq V$ is a subset of vertices that have already been included in the solution. For a vertex $a \in V \backslash S$ let $Q(G, S, a) = \max_{S' \supseteq S \cup \{a\}, S' \in \mathcal{F}} f(S') - f(S \cup \{a\})$ denote the marginal utility gained by selecting vertex $a$. Such a *Q-function* satisfies the Bellman equations given by

$$
\begin{aligned}
Q(G, S, a) =& f(S \cup \{a\}) - f(S) \\
& + \max_{a' \in V \backslash S \cup \{a\}} Q(G, S \cup \{a\}, a').
\end{aligned}
\tag{18}
$$

It is easily seen that computing the $Q$-functions solves the optimization problem, as $\max_{S \in \mathcal{F}} f(S) = \max_{a \in V} Q(G, \{\}, a)$. However exactly computing $Q$-functions may be computationally expensive. One approach towards approximately computing $Q(G, S, a)$ is to fit it to a (polynomial time computable) parametrized function, in a way that an appropriately defined error metric is minimized. This approach is called *Q-learning* in the reinforcement learning (RL) paradigm, and is described below.

**State, action and reward.** We consider a reinforcement learning policy in which the solution set $S \subseteq V$ is generated one vertex at a time. The algorithm proceeds in rounds, where at round $t$ the RL agent is presented with the graph $G$ and the set of vertices $S_t$ chosen so far. Based on this *state* information, the RL agent outputs an *action* $A_t \in V \backslash S_t$. The set of selected vertices is updated as $S_{t+1} = S_t \cup \{A_t\}$. Initially $S_0 = \{\}$. Every time the RL agent performs an action $A_t$ it also incurs a *reward* $R_t = f(S_t \cup \{A_t\}) - f(S_t)$. Note that the $Q$-function $Q(G, S_t, a)$ is well-defined only if $S_t$ and $a$ are such that there exists an $S' \supseteq S_t \cup \{a\}$ and $S' \in \mathcal{F}$. To enforce this let $\mathcal{F}_t = \{a \in V \backslash S_t : \exists S' \text{ s.t. } S' \supseteq S_t \cup \{a\} \text{ and } S' \in \mathcal{F}\}$ denote the set of feasible actions at time $t$. Each round, the learning agent chooses an action $A_t \in \mathcal{F}_t$. The algorithm terminates when $\mathcal{F}_t = \{\}$.

**Policy.** The goal of the RL agent is to learn a *policy* for selecting actions at each time, such that the cumulative reward incurred $\sum_{t \geq 0} R_t$ is maximized. A measure of the generalization capability of the policy is how well it is able to maximize cumulative reward for different graph instances from a collection (or from a distribution) of interest.

**$Q$-learning.** Let $\tilde{Q}(G, S, a; \Theta)$ denote the approximation of $Q(G, S, a)$ obtained using a parametrized function with parameters $\Theta$. Further let $((G_i, S_i, a_i))_{i=1}^N$ denote a sequence of (state,

action) tuples available as training examples. We define empirical loss as

$$\hat{L} = \sum_{i=1}^{N} \left( \tilde{Q}(G_i, S_i, a_i; \Theta) - f(S_i \cup \{a_i\}) + f(S_i) \right.$$

$$\left. - \max_{a' \in V \setminus S_i \cup \{a_i\}} \tilde{Q}(G_i, S_i \cup \{a_i\}, a'; \Theta) \right)^2, \quad (19)$$

and minimize using stochastic gradient descent. The solution of the Bellman equations equation 18 is a stationary point for this optimization.

**Remark.** Heuristics such as ours, which select vertices one at a time in an irreversible fashion are studied as 'priority greedy' algorithms in computer science literature (Borodin et al., 2003; Angelopoulos & Borodin, 2003). The fundamental limits (worst-case) of priority greedy algorithms for minimum vertex cover and maximum independent set has been discussed in Borodin et al. (2010).

## C.2   SEQ2VEC UPDATE EQUATIONS

**Seq2Vec.** The sequence $(\{\mathbf{x}_v(t) : v \in V\})_{t=1}^{T}$ is processed by a gated recurrent network that sequentially reads $\mathbf{x}_v(\cdot)$ vectors at each time index for all $v \in V$. Standard GRU (Chung et al., 2014). For time-step $t \in \{1, \ldots, T\}$, let $\mathbf{y}_v(t) \in \mathbb{R}^d$ be the $d$-dimensional cell state, $\mathbf{i}_v(t) \in \mathbb{R}^d$ be the cell input and $\mathbf{f}_v(t) \in (0,1)^d$ be the forgetting gate, for each vertex $v \in V$. Each time-step a fresh input $\mathbf{i}_v(t)$ is computed based on the current states $\mathbf{x}_u(t)$ of $v$'s neighbors in $G$. The cell state is updated as a convex combination of the freshly computed inputs $\mathbf{i}_v(t)$ and the previous cell state $\mathbf{y}_v(t-1)$, where the weighting is done according to a forgetting value $\mathbf{f}_v(t)$ that is also computed based on the current vertex states. The update equations for the input vector, forgetting value and cell state are chosen as follows:

$$\mathbf{i}_v(t+1) = \text{relu}(\mathbf{W}_4 \sum_{u \in \Gamma(v)} \mathbf{x}_v(t) + \mathbf{w}_5 c_v(t) + \mathbf{b}_6)$$

$$\mathbf{f}_v(t+1) = \text{sigmoid}(\mathbf{W}_7 \sum_{u \in V} \mathbf{x}_u(t) + \mathbf{b}_8)$$

$$\mathbf{y}_v(t+1) = \mathbf{f}_v(t+1) \odot \mathbf{i}_v(t+1) + (\mathbf{1} - \mathbf{f}_v(t+1)) \odot \mathbf{y}_v(t), \quad (20)$$

where $\mathbf{W}_4, \mathbf{W}_7 \in \mathbb{R}^{d \times d}$ and $\mathbf{w}_5, \mathbf{b}_6, \mathbf{b}_8 \in \mathbb{R}^d$ are trainable parameters, $t = 0, 1, \ldots, T-1$, and $\mathbf{1}$ denotes the $d$-dimensional all-ones vector, and $\odot$ is element-wise multiplication. $\mathbf{y}_v(0)$ is initialized to all-zeros for every $v \in V$. The cell state at the final time-step $\mathbf{y}_v(T), v \in V$ is the desired vector summary of the GRAPH2SEQ sequence.

## C.3   TIME-COMPLEXITY

The time-complexity of testing a graph $G(V, E)$ with $|V|$ nodes, $|E|$ edges using Algorithm 2 is $O((|E| + |V|)T_{\max}^2|V|)$. This can be derived as follows. The time-complexity for one forward pass of G2S-RNN (e.g., to select one vertex in minimum vertex cover) is $O((|E| + |V|)T_{\max})$. This is because during each step of GRAPH2SEQ, $O(|E|)$ operations are required to update the state at each vertex based on neighbors' states, and $O(|V|)$ operations are required by the GRU to aggregate the states of all vertices (equation 20). Since these operations have to be repeated at each step, and there are $T_{\max}$ steps, the time-complexity is $O((|E| + |V|)T_{\max})$.

Now, for a fixed number of steps $T$, the time-complexity to compute a complete solution (e.g., to select multiple vertices such that they form a valid vertex cover) is $O((|E| + |V|)T_{\max} * |V|)$. This is because selecting one vertex has complexity $O((|E| + |V|)T_{\max})$, and we may have to select $O(|V|)$ vertices to obtain a valid solution to the input graph.

Finally, the overall time-complexity is $O((|E| + |V|)T_{\max} * |V| * T_{\max})$. This is because the final solution is computed by first computing valid solutions for each $T = 1, 2, \ldots, T_{\max}$, and then selecting the best valid solution from among them. Computing a valid solution for a fixed $T$ takes $O((|E| + |V|)T_{\max} * |V|)$ as mentioned above, and we have to repeat the process $T_{\max}$ times.

Note that aggregating states from all the vertices in the GRU is a hyperparameter choice. If only local neighborhood states are used in the GRU, the time-complexity of one forward pass in the above becomes $O(|E|T_{\max} + |V|)$.

| Size \ Sch | G2S | S2V | S2V-dyn | S-GCN | S-mean | S-pool | WL-NN | WL | List | Match | Greedy | Gurobi |
|---|---|---|---|---|---|---|---|---|---|---|---|---|
| 25 | 0.384 | 0.011 | 0.152 | 0.004 | 0.006 | 0.008 | 0.009 | 0.051 | 2.0e-4 | 6.2e-6 | 1.1e-5 | 8.3e-5 |
| 50 | 0.372 | 0.010 | 0.158 | 0.004 | 0.005 | 0.009 | 0.010 | 0.724 | 9.0e-4 | 6.1e-5 | 4.3e-5 | 8.0e-4 |
| 100 | 0.432 | 0.013 | 0.190 | 0.006 | 0.006 | 0.015 | 0.011 | 12.50 | 7.0e-4 | 1.0e-4 | 2.0e-4 | 0.007 |
| 200 | 0.714 | 0.027 | 2.508 | 0.015 | 0.018 | 0.068 | 0.017 | — | 0.001 | 7.0e-4 | 0.001 | 4.912 |
| 400 | 2.34 | 0.127 | 5.148 | 0.057 | 0.061 | 0.416 | 0.048 | — | 0.004 | 0.008 | 0.009 | OOM |
| 800 | 16.11 | 0.847 | 11.05 | 0.282 | 0.320 | 3.035 | 0.205 | — | 0.009 | 0.057 | 0.061 | OOM |
| 1600 | 108.1 | 6.203 | 56.62 | 2.096 | 2.418 | 24.03 | 1.433 | — | 0.030 | 0.530 | 0.516 | OOM |
| 3200 | 641.6 | 43.97 | 433.5 | 14.86 | 17.43 | 192.3 | 11.04 | — | 0.101 | 4.792 | 4.723 | OOM |

Table 1: Running time (in minutes) of G2S-RNN and baseline heuristics for minimum vertex cover on random Erdos-Renyi graphs. The WL heuristic could not be run on graphs of size greater than 200 due the large number of serial computations involved which makes it prohibitively slow. OOM refers to out-of-memory.

## D  EVALUATION DETAILS

### D.1  HEURISTICS COMPARED

We compare G2S-RNN against:

(1) *Structure2Vec* (Khalil et al., 2017), a spatial GCNN with depth of 5.
(2) *GraphSAGE* (Hamilton et al., 2017a) using (a) GCN, (b) mean and (c) pool aggregators, with the depth restricted to 2 in each case.
(3) *WL kernel NN* (Lei et al., 2017), a neural architecture that embeds the WL graph kernel, with a depth of 3 and width of 4 (see Lei et al. (2017) for details).
(4) *WL kernel embedding*, in which the feature map corresponding to WL subtree kernel of the subgraph in a 5-hop neighborhood around each vertex is used as its vertex embedding (Shervashidze et al., 2011). Since we test on large graphs, instead of using a learned label lookup dictionary we use a standard SHA hash function for label shortening at each step. In each of the above models, the outputs of the last layer are fed to a $Q$-learning network, and trained the same way as G2S-RNN.
(5) *S2V-dynamic*, a variant of Structure2Vec (Khalil et al., 2017) in which the number of convolution steps $T$ is varied from 1 to 15. For testing an input graph instance $G$, we first compute a separate solution for each fixed $T$ in the specified range; the final solution is selected as the best solution among the 15 solutions. The procedure is analogous to Algorithm 2. While training the number of convolution steps is set to $T = 5$.
(6) *Greedy algorithms.* We consider greedy heuristics (Williamson & Shmoys, 2011) for each of MVC, MC and MIS.
(7) *List heuristic.* A fast list-based algorithm proposed recently in Shimizu et al. (2016) for MVC and MIS.
(8) *Matching heuristic.* A 2-approximation algorithm for MVC (Williamson & Shmoys, 2011).

### D.2  TRAINING DETAILS

We train G2S-RNN, Structure2Vec, S2V-dynamic, GraphSAGE and WL kernel NN all on small Erdos-Renyi (ER) graphs of size 15, and edge probability 0.15. During training we truncate the G2S-RNN to 5 steps (i.e., $T = 5$, see Fig. 1). In each case, the model is trained for 100,000 iterations, except WL kernel NN which is trained for 200,000 iterations since it has more parameters. We use experience replay (Mnih et al., 2013), a learning rate of $10^{-3}$, Adam optimizer (Kingma & Ba, 2015) and an exploration probability that is reduced from 1.0 to a resting value of 0.05 over 10,000 iterations. The amount of noise added in the evolution ($n_v(t)$ in Equation 1) seemed to not matter; we have set the noise variance $\sigma^2$ to zero in all our experiments (training and testing). As far as possible, we have tried to keep the hyperparameters in G2S-RNN and all the neural network baselines to be the same. For example, all of the networks have a vertex embedding dimension of 16, use the same neural architecture for the $Q$-network and Adam optimizer for training.

### D.3  RUNNING TIME

Table 1 presents the average run times of G2S-RNN and baseline schemes for the minimum vertex cover problem on random Erdos-Renyi graphs ($p = 0.15$). The experiments were run on a c4.4x

| Size \ Sch | G2S | S2V | S2V-dyn | S-GCN | S-mean | S-pool | WL-NN | WL | Greedy | Gurobi |
|---|---|---|---|---|---|---|---|---|---|---|
| 25 | 0.271 | 0.001 | 0.099 | 0.004 | 0.005 | 0.008 | 0.009 | 0.042 | 8.2e-6 | 3.0e-4 |
| 50 | 0.275 | 0.001 | 0.102 | 0.004 | 0.005 | 0.008 | 0.010 | 0.487 | 2.7e-5 | 0.009 |
| 100 | 0.293 | 0.002 | 0.117 | 0.005 | 0.006 | 0.012 | 0.011 | 8.048 | 9.8e-5 | 13.34 |
| 200 | 0.362 | 0.016 | 0.009 | 0.009 | 0.010 | 0.046 | 0.015 | 168.7 | 4.0e-4 | > 2 days |
| 400 | 0.730 | 0.033 | 0.548 | 0.035 | 0.034 | 0.254 | 0.039 | — | 0.002 | > 2 days |
| 800 | 3.569 | 0.239 | 3.416 | 0.270 | 0.273 | 1.853 | 0.223 | — | 0.011 | > 2 days |
| 1600 | 22.68 | 1.609 | 22.40 | 1.705 | 1.771 | 14.03 | 1.558 | — | 0.065 | > 2 days |
| 3200 | 192.6 | 14.10 | 191.2 | 14.46 | 15.24 | 111.3 | 13.66 | — | 0.493 | > 2 days |

Table 2: Running time (in minutes) of G2S-RNN and baseline heuristics for max cut on random Erdos-Renyi graphs.

| Size \ Sch | G2S | S2V | S2V-dyn | S-GCN | S-mean | S-pool | WL-NN | WL | List | Greedy | Gurobi |
|---|---|---|---|---|---|---|---|---|---|---|---|
| 25 | 0.348 | 0.001 | 0.097 | 0.004 | 0.005 | 0.008 | 0.010 | 0.033 | 2.0e-4 | 1.1e-5 | 3.8e-5 |
| 50 | 0.345 | 0.001 | 0.098 | 0.004 | 0.005 | 0.008 | 0.010 | 0.232 | 3.0e-4 | 2.7e-5 | 2.0e-4 |
| 100 | 0.353 | 0.001 | 0.099 | 0.004 | 0.005 | 0.009 | 0.010 | 2.543 | 6.0e-4 | 8.0e-5 | 0.004 |
| 200 | 0.369 | 0.001 | 0.106 | 0.005 | 0.006 | 0.015 | 0.010 | 33.24 | 0.001 | 2.0e-4 | 4.339 |
| 400 | 0.426 | 0.002 | 0.125 | 0.006 | 0.007 | 0.036 | 0.012 | — | 0.003 | 9.0e-4 | OOM |
| 800 | 0.657 | 0.006 | 0.177 | 0.013 | 0.014 | 0.129 | 0.015 | — | 0.010 | 0.004 | OOM |
| 1600 | 1.654 | 0.019 | 0.363 | 0.035 | 0.036 | 0.525 | 0.031 | — | 0.033 | 0.017 | OOM |
| 3200 | 5.948 | 0.072 | 1.216 | 0.132 | 0.140 | 2.255 | 0.108 | — | 0.131 | 0.081 | OOM |

Table 3: Running time (in minutes) of G2S-RNN and baseline heuristics for maximum independent set on random Erdos-Renyi graphs.

large instance with 16 vCPUs and 30 GB of memory. For each scheme, except the WL algorithm, the graph size is varied from 25–3200 as in §5. The large number of serial computations in the WL algorithm makes it prohibitively slow on large graphs and hence was not evaluated. We notice that on small graphs (size < 200), the Gurobi solver is faster than G2S-RNN. At graph size 200, Gurobi becomes $8\times$ slower than G2S-RNN. For graph sizes greater than 200, the Gurobi solver could not compute a solution even after 4 days and ran out of memory. We also notice that on large graphs G2S-RNN has run times that are roughly $15\times$ that of Structure2Vec. This is as expected since for a fixed $T$, G2S-RNN($T$) has complexity similar to Structure2Vec, and we are evaluating it at each $T = 1, 2, \ldots, 15$.

Tables 2 and 3 show the run times for the max cut and maximum independent set problems respectively, on random Erdos-Renyi graphs ($p = 0.15$). As in the case of minimum vertex cover, we observe the Gurobi solver to be faster than G2S-RNN on small graphs. At larger graph sizes, e.g., size 100 in MC and size 200 in MIS, Gurobi is $45\times$ and $11\times$ slower respectively than G2S-RNN. At even larger graph sizes, Gurobi is either unable to compute a solution even after two days (MC) or runs out of memory (MIS).

## D.4 ADVERSARIAL TRAINING

So far we have seen the generalization capabilities of a G2S-RNN model trained on small Erdos-Renyi graphs. In this section we ask the question: is even better generalization possible by training on a different graph family? The answer is in the affirmative. We show that by training on *planted vertex-cover* graph examples—a class of 'hard' instances for MVC—we can realize further generalizations. A planted-cover example is a graph, in which a small graph is embedded ('planted') within a larger graph such that the vertices of the planted graph constitute the optimal minimum vertex cover. Figure 7 shows the result of testing G2S-RNN models trained under both Erdos-Renyi and planted vertex cover graphs. While both models show good scalability in Erdos-Renyi and regular graphs, on bipartite graphs and worst-case graphs the model trained on planted-cover graphs shows even stronger consistency by staying 1% within optimal.

## D.5 GEOMETRY OF ENCODING AND SEMANTICS OF GRAPH2SEQ

Towards an understanding of what aspect of solving the MVC is learnt by GRAPH2SEQ, we conduct empirical studies on the dynamics of the state vectors as well as present techniques and semantic interpretations of GRAPH2SEQ.

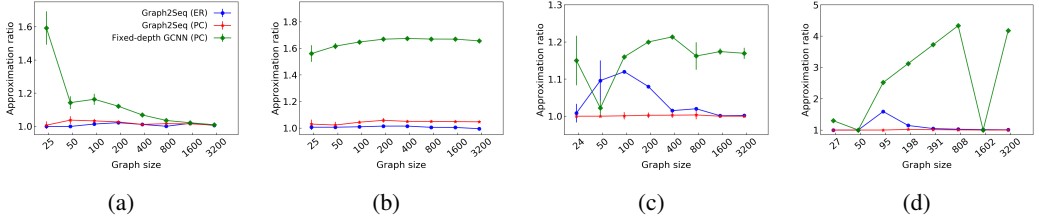

Figure 7: Minimum vertex cover in (a) random Erdos-Renyi graphs, (b) random regular graphs, (c) random bipartite graphs, (d) greedy example, under Erdos-Renyi graph and adversarial graph training strategies.

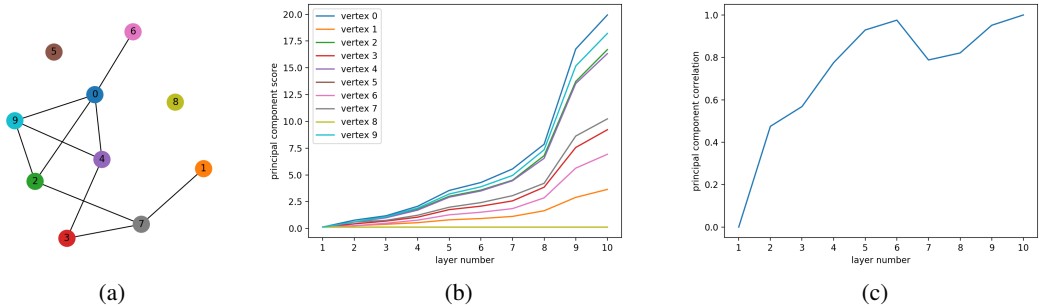

Figure 8: (a) Erdos-Renyi graph of size 10 considered in Figures (b) and (c), (b) vertex-wise principal component scores at each layer, and (c) projection of the principal direction at each iteration on the principal direction of iteration 10. These experiments are performed on our trained model.

In the first set of experiments, we investigate the vertex state vector sequence. We consider graphs of size up to 50 and of types discussed in Section 5. For each fixed graph, we observe the vertex state $\mathbf{x}(\cdot)$ (Equation 1) evolution to a depth of 10 layers.

**(1) Dimension collapse.** As in the random parameter case, we observe that on an average more than 8 of the 16 dimensions of the vertex state become zeroed out after 4 or 5 layers.

**(2) Principal components' alignment.** The principal component direction of the vertex state vectors at each layer converges. Fig. 8(c) shows this effect for the graph shown in Fig. 8(a). We plot the absolute value of the inner product between the principal component direction at each layer and the principal component direction at layer 10.

**(3) Principal component scores and local connectivity.** The component of the vertex state vectors along the principal direction roughly correlate to how well the vertex is connected to the rest of the graph. We demonstrate this again for the graph shown in Fig. 8(a), in Fig 8(b).

**(4) Optimal depth**. We study the effect of depth on approximation quality on the four graph types being tested (with size 50); we plot the vertex cover quality as returned by GRAPH2SEQ as we vary the number of layers up to 25. Fig. 9(a) plots the results of this experiment, where there is no convergence behavior but nevertheless apparent that different graphs work optimally at different layer values. While the optimal layer value is 4 or 5 for random bipartite and random regular graphs, the worst case greedy example requires 15 rounds. This experiment underscores the importance of having a *flexible* number of layers is better than a fixed number; this is only enabled by the time-series nature of GRAPH2SEQ and is inherently missed by the fixed-depth GCNN representations in the literature.

**(5) $Q$-function semantics.** Recall that the $Q$-function of equation 2 comprises of two terms. The first term, denoted by $Q_1$, is the same for all the vertices and includes a sum of all the $\mathbf{y}(\cdot)$ vectors. The second term, denoted by $Q_2(v)$ depends on the $\mathbf{y}(\cdot)$ vector for the vertex being considered. In this experiment we plot these two values at the very first layer of the learning algorithm (on a planted vertex cover graph of size 15, same type as in the training set) and make the following observations: (a) the values of $Q_1$ and $Q_2(\cdot)$ are close to being integers. $Q_1$ has a value that is one less than the

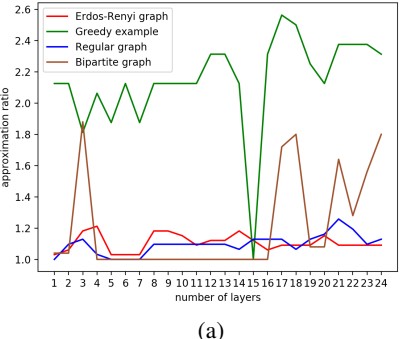 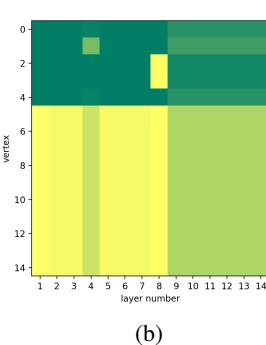 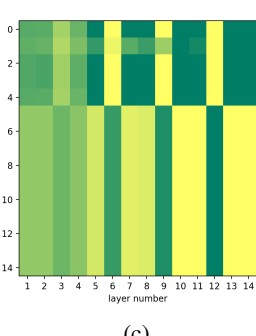

(a)             (b)             (c)

Figure 9: (a) Approximation ratio of GRAPH2SEQ with varying number of layers, (b) $\mathbf{y}(\cdot)$ vectors of GRAPH2SEQ in the intermediate layers seen using the $Q$-function, (c) $\mathbf{x}(\cdot)$ vectors of the fixed-depth model seen using the $Q$-function. Figure (b) and (c) are on planted vertex cover graph with optimum cover of vertices $\{0, 1, 2, 3, 4\}$.

negative of the minimum vertex cover size. (b) For a vertex $v$, $Q_2(v)$ is binary valued from the set $\{0, 1\}$. $Q_2(v)$ is one, if vertex $v$ is part of an optimum vertex cover, and zero otherwise. Thus the neural network, in principle, computes the complete set of vertices in the optimum cover at the very first round itself.

**(6) Visualizing the learning dynamics**. The above observations suggests to 'visualize' how our learning algorithm proceeds in each layer of the evolution using the lens of the value of $Q_2(\cdot)$. In this experiment, we consider size-15 planted vertex cover graphs on (i) GRAPH2SEQ, and (ii) the fixed-depth GCNN trained on planted vertex cover graphs. Fig. 9(b) and 9(c) show the results of this experiment. The planted vertex cover graph considered for these figures has an optimal vertex cover comprising vertices $\{0, 1, 2, 3, 4\}$. We center (subtract mean) the $Q_2(\cdot)$ values at each layer, and threshold them to create the visualization. A dark green color signifies the vertex has a high $Q_2(\cdot)$ value, while the yellow means a low $Q_2(\cdot)$ value. We can see that in GRAPH2SEQ the heuristic is able to compute the optimal cover, and moreover this answer does not change with more rounds. The fixed depth GCNN has a non-convergent answer which oscillates between a complementary set of vertices. Take away message: having an upper LSTM layer in the learning network is critical to identify when an optimal solution is reached in the evolution, and "latch on" to it.

