# OpenReview forum: "Graph2Seq: Scalable Learning Dynamics for Graphs"
_ICLR.cc/2019/Conference_

### Official Review · AnonReviewer2 · 2018-10-31
**Interesting idea with weaknesses in the formal and empirical parts**

**Rating:** 4
**Confidence:** 4

**Review:**

The authors propose a method for learning vector representations for graphs. The problem is relevant to the ICLR community.

The paper has, however, three major problems:

The motivation of the paper is somewhat lacking. I agree that learning representations for graphs is a very important research theme. However, the authors miss to motivate their specific approach. They mention the importance of learning on smaller graphs and applying the learned models to larger graphs (i.e., extrapolating better). I would encourage the authors to elaborate on some use cases where this is important. I cannot think of any at the moment. I assume the authors had use cases in combinatorial optimization in mind? Perhaps it might make sense to motivate the use of GNNs to solve vertex cover etc.

I’m not sure about the correctness of some of the theorems. For instance, Theorem 2 states
“For any fixed k > 0, there exists a function f(·) and an input graph instance G such that no k-LOCAL-GATHER algorithm can compute f(G) exactly.”  I’m not claiming that this is a false statement. What I am suspecting at the moment is that the proof might not necessarily be correct. For instance, it is known that what you call 1-LOCAL-GATHER can compute the 1-Weisfeiler-Leman partition of the nodes (sometimes also referred to as the 1-WL node coloring). Now consider the chain graph 1 - 2 - 3 - 4 - 5. Here, the partition that puts together 1-WL indistinguishable nodes are {1, 5}, {2, 4} and {3}. Hence, the 1-WL coloring is able to distinguish say nodes 2 and 3 even their 1-neighborhood looks exactly the same. A similar argument might apply to your example pairs of graphs but I haven’t checked it yet in detail. What is for sure though: what you provide in the appendix is not a proper formal proof of Theorem 2. This has to be fixed.

The experiments are insufficient. The authors should compare to existing methods on common benchmark problems such as node or graph classification datasets. Comparing to baselines on a new set of task is not enough. Why not compare your method also on existing datasets?
If you motivate your method as one that performs well on combinatorial problems (e.g., vertex cover) you should compare to existing deterministic solvers. I assume that these are often much faster at least on smaller graphs.

---

> ### Author Response · Authors · 2018-11-15
> **Author response**
>
> Thank you for the helpful comments.
>
> 1. Motivation: Our primary motivation is learning algorithms for combinatorial optimization on graphs. In many practical applications, it is desirable to learn an algorithm on smaller graphs that can generalize to larger graphs. For example, Mirrhosseini et al. [1] consider the problem of deciding how to optimally assign TensorFlow ops to GPUs to minimize runtime. Since directly training placement policies on large TensorFlow graphs can be extremely slow, it would be beneficial if a model can be trained on small TensorFlow graphs in a way that generalizes to large TensorFlow graphs. Another example is query optimization in databases, where the optimal order of join operations in the query plan tree has to be determined [2]. Since evaluating complex queries with large query plans can be expensive, it is again helpful if the learning algorithm can be trained on simple queries in a way that generalizes to complex queries. We will modify the introduction to emphasize these use cases.
>
> 2. Theorem 2: The chain graph 1-2-3-4-5 results in the partitions {1, 5}, {2, 4} and {3} after the 1-hop WL algorithm if the initial node labels are chosen as their respective degrees. Since the degree label already includes one-hop information, this means, overall it is a 2-local-gather algorithm and not a 1-local-gather algorithm. If the initial node labels are chosen identically for all nodes, then the partitions are {1, 5}, {2, 3, 4}. We would appreciate clarification on what parts of the proof of Theorem 2 are unclear or informal.
>
> 3. Baselines: As our focus is on combinatorial optimization problems, comparing on benchmark node or graph classification datasets is outside the scope of this paper and is an important future research direction. We have compared Graph2Seq to existing deterministic solvers (Gurobi), heuristics (list), approximation algorithms (matching, greedy) and a range of graph neural networks. Note that the performance plots for the Gurobi solver is implicit in the plots (e.g., Figure 2 and 3) since the approximation ratio for all other schemes have been computed relative to the Gurobi solver. The plots corresponding to the list heuristic (brown), matching algorithm (green) and greedy heuristic (yellow) are explicitly shown in Figures 2 and 3.
>
> [1] Device placement optimization with reinforcement learning, Mirhoseini et al, 2017
> [2] Learning to optimize join queries with deep reinforcement learning, Krishnan et al, 2018

---

> > ### Comment · AnonReviewer2 · 2018-11-15
> > **Misunderstandings**
> >
> > Dear authors,
> >
> > Let me quickly clarify some misunderstandings.
> >
> > Theorem 2:
> >
> > - The statement "the degree label already includes one-hop information" is not correct. One-hop means considering information that is up to one hop away. The degree of the node itself is "0"-hops away.
> > - In your proof, you chose k=2 and then show two graphs of which you claim that there are nodes that have the same 2-hop neighborhood and, therefore, cannot be distinguished by a 2-local-gather approach. What I have tried to point out is that even if two nodes have the same neighborhood (for k = 1), they can be distinguished with 1-local-gather. The argument can be extended for larger k as well. Just increase the size of the chain: A-B-C-D-E-F-G. Now, C and D have the same 2-hop neighborhood and *still* can be distinguished with 1-WL which is a 1-local-gather approach (even if it is a 2-local-gather approach, as you claim, which is not correct because 1-WL only looks at the 1-hop neighborhood). What you are missing is that an algorithm that "only" looks at the k-hop neighborhood can still distinguish two nodes with identical k-hop neighborhood (but different l-neighborhood, l>k) due to the iterative (i.e., recursive) application of the algorithm. The information is propagated throughout the graph. That's what the 1-WL algorithm is doing.
> > - You proof for k=2 (which is not correct as pointed out above) and write: "the example easily generalizes for larger k". You would have to at least make an argument how you want to generalize.
> >
> > Overall, there seem to be several gaps in your proof. Again, the statement "a k-local-gather approach cannot distinguish nodes in the graph that have the same k-hop neighborhood" is incorrect.
> >
> > Baselines:
> >
> > Since your motivation seems to be solving combinatorial optimization problems with a graph NN and since according to your response you have used Gurobi, why not report the running times? The main problem one is faced when working with combinatorial optimization problems is the time it takes to solve them. Even if you don't stand a chance to be faster than Gurobi on smaller problems, you should be able to generate larger problems and, eventually, there should be a break-even point where your method is faster than Gurobi. It is these types of investigations that I am completely missing in the paper. There is not a single mention of the time it took to run the baselines, Gurobi, or your proposed method. (Please correct me if I'm wrong.)
> > Sure, it is interesting that you can approximate the solutions more tightly but at what cost? If your method takes always longer than an optimal solver, what's the point?
> >
> > Concerning experiments on other benchmark datasets: This is not strictly necessary. I agree. But it would make the paper much stronger.

---

> > > ### Author Response · Authors · 2018-11-16
> > > **Author response 1**
> > >
> > > Thank you for the comments.
> > >
> > > 1. Clarification on node label:
> > > We first note that node labels greatly affect how many local aggregation steps are required to compute a function on a graph. For example, if nodes are labeled by their degrees then, as you have correctly pointed out, a 1-local algorithm is enough to distinguish vertices in the example of line graphs such as 1-2-3-4-5. In general, having node degree as a label can allow vertex embeddings in a k-local algorithm to depend on the (k+1)-hop neighborhood around the vertices. In the extreme case, node labels can encode the entire structure of the graph, e.g., if the adjacency matrix is used as a label. In this case, even a 0-local algorithm will be able to exactly compute any function over the graph because the entire graph can be inferred just by looking at a node's label. The graphs we consider in the proof of Theorem 2 are unlabeled graphs. In unlabeled graphs, attributes such as node degree are not inherently part of a node and need to be explicitly computed.
> > >
> > > WL being 1-local vs 2-local:
> > > We note that Theorem 2 makes an existential statement: for a fixed k, we are saying there *exists* some graph G and some function f(.) such that f(G) cannot be computed exactly by any k-local-gather algorithm. Therefore, to prove the theorem, all we need is one example graph G and one example function f(.) where f(G) cannot be computed exactly by any k-local-gather algorithm. We have freedom to select whatever G and f(.) we want, as long as we can prove f(G) cannot be computed exactly by any k-local-gather algorithm.
> > >
> > > The graph G that we choose in our proof is an *unlabeled* tree shown in Figure 5, without any node or edge attributes or features. If we run a 1-hop WL on such a graph, the first step will be to assign the respective degrees of nodes as the starting color. However, the graph we have chosen is such that there are no features intrinsically present within each node. In particular, the node degree information is absent in the nodes, and hence need to be explicitly computed.
> > >
> > > If we consider the 0-hop neighborhood of each node---basically just the nodes themselves---then we cannot compute the node degree since a node in isolation does not reveal how many nodes it is connected to. Therefore, we must look at the 1-hop neighborhood---the node together with all the nodes it is connected to---in order to compute the degree. Once the initial color (node degree) has been computed, we can then proceed to do one step of neighborhood color aggregation as per the WL algorithm and compute the updated node colors. Thus overall, we have performed two steps of 1-hop aggregation operations---one for computing the node degrees, and one for aggregating neighboring colors---which makes the 1-hop WL algorithm a 2-local-gather algorithm in our chosen graph.
> > >
> > > For some other choice of the graph, the 1-hop WL algorithm will be a 1-local-gather algorithm. For example, we can choose the graph to be such that each node intrinsically includes its degree as a label. In such a graph, the initial node degree color can be computed by looking at the 0-hop neighborhood of nodes, since the node itself (in isolation) includes this information; we do not need to look at the 1-hop neighborhood.
> > >
> > > Clarification of definition of local-gather:
> > > A k-local-gather algorithm, according to our definition, consists of two steps: a local step which computes an embedding for each vertex based on its k-hop neighborhood, and a gather step which aggregates embeddings from all vertices. In particular, the local step occurs only once, and is not computed k times. Therefore, for unlabeled feature-less graphs, such as the one we have considered in the proof, information from outside the k-hop neighborhood of a vertex cannot reach the vertex in just one step of local computation. Consequently, a k-hop local-gather algorithm cannot distinguish vertices having identical k-hop neighborhood even if the l-hop neighborhoods (for l > k) are different.
> > >
> > > Algorithms such as the k-WL or k-step GCNN perform 1-hop local neighborhood aggregation repeatedly for k (or k+1) steps. However, these algorithms are still valid k-local-gather (or (k+1)-local-gather) algorithms. This is because: (i) the k (or k+1) steps of 1-hops aggregation causes each vertex embedding to depend on its k (or k+1) hop neighborhood, and (ii) we can think of the k (or k+1) steps of 1-hop aggregation in these algorithms together as forming the local step of a k-local-gather (or (k+1)-local-gather) algorithm. This is why a 1-hop WL algorithm which performs two steps of 1-hop aggregation operations is a valid 2-local-gather algorithm.

---

> > > > ### Author Response · Authors · 2018-11-16
> > > > **Author response 2**
> > > >
> > > > Extending proof to larger k:
> > > > The proof of Theorem 2 can easily be extended to larger k by considering the same two trees in Figure 5, but with longer chains around each degree-3 node. Currently, around each degree-3 node there are chains of nodes, with three nodes in each chain. For a general k, we would increase the chains of three nodes to chains of l nodes, where l is any odd number greater than k. This has already been mentioned in the proof. The proof itself is identical. We will elaborate it further in the revision.
> > > >
> > > > We realize these can be confusing points and thank you for bringing this to attention. We will include explanations in the paper to clarify node labeling and the other points. We hope this answer clarifies any misunderstandings.
> > > >
> > > > 2. Run times: For large graphs (size > 400) we indeed observed that Gurobi is much slower and took many hours or days. We will include measurement data on how long it takes for Gurobi and other baselines to compute a solution in our revision.

---

> > > > > ### Comment · AnonReviewer2 · 2018-12-03
> > > > > **Discussion**
> > > > >
> > > > > Dear authors,
> > > > >
> > > > > Thanks for the further responses. I agree with all of your statements (which do not contradict what I said in my response) until this part:
> > > > >
> > > > > "A k-local-gather algorithm, according to our definition, consists of two steps: a local step which computes an embedding for each vertex based on its k-hop neighborhood, and a gather step which aggregates embeddings from all vertices. In particular, the local step occurs only once, and is not computed k times. Therefore, for unlabeled feature-less graphs, such as the one we have considered in the proof, information from outside the k-hop neighborhood of a vertex cannot reach the vertex in just one step of local computation. Consequently, a k-hop local-gather algorithm cannot distinguish vertices having identical k-hop neighborhood even if the l-hop neighborhoods (for l > k) are different."
> > > > >
> > > > > But this definition of k-local-gather is problematic. For instance, the Kipf and Welling GCN is applied in a way that the local (aggregation) steps do *not* occur only once. That's why this method is able to propagate information throughout the graph. Again, GCNs do not just apply the local step once. Therefore, your statement that GCNs are 4-local-gather (with the definition you mention here) is incorrect.
> > > > >
> > > > > You cannot have it both ways.
> > > > >
> > > > > Either you define the "k" in local-gather has the distance to the currently considered node used to aggregate neighborhood information *in every learning step*. Then my argument above that 1-WL is 2-local-gather (if no node degrees are available on the nodes and otherwise 1-local-gather) holds. And my argument still holds that your proof is not correct since you prove for k=2.
> > > > >
> > > > > Or you define the "k" in local-gather as you wrote in the response above. You only apply the k-hop aggregation once. But then statements such as those made in Proposition 2 cannot be correct. Then GCNs are not 4-local-gather since you are not applying the neighborhood aggregation step only once.
> > > > >
> > > > > Due to these inconsistencies which are caused by inconsistent/ambiguous definitions in the best case and by incorrect arguments in the worst case, I will keep the original score. The paper has improved especially since you added additional timing results but the issues above are severe enough that I don't have enough confidence in the correctness of the statements made in the paper.

---

> > > > > > ### Comment · AnonReviewer1 · 2018-12-04
> > > > > > **Agreed**
> > > > > >
> > > > > > Thanks to the authors and Reviewer 2 for discussing this point in detail.
> > > > > >
> > > > > > I strongly agree with Reviewer 2 on this issue, and that is why I asked about the 4-local-gather in my original review.
> > > > > >
> > > > > > I too cannot see how the Graph2Seq model is infty-local-gather but the Kipf-GCN or Structure2Vec are not, given that both are iterative algorithms that can aggregate information from all nodes if run for diameter-iterations.

---

> > > > > > > ### Author Response · Authors · 2018-12-08
> > > > > > > **Author response**
> > > > > > >
> > > > > > > GCN algorithms including Kipf and Welling, Structure2Vec etc., all perform a fixed number of local steps (e.g., 4), regardless of the size of the input graph. We are not aware of any work where these algorithms dynamically adjust the number of local steps based on the input graph instance.
> > > > > > >
> > > > > > > The point of our work is that dynamically adjusting the number of local steps, results in algorithms that generalize and scale better. In addition, we argue that considering outputs of all intermediate local steps (i.e., the sequence of local step outputs) in the aggregation step, results in even better generalization.

---

> > > > > > ### Author Response · Authors · 2018-12-08
> > > > > > **Author response 1**
> > > > > >
> > > > > > Thank you for highlighting this cause for confusion.
> > > > > >
> > > > > > We define a k-local-gather algorithm as a two-phase algorithm comprising of a local phase, and a gather phase. In the local phase each vertex computes a representation based on its k-hop neighborhood subgraph. In the gather phase, the vertex representations are aggregated to compute the final output of the algorithm.
> > > > > >
> > > > > > GCN algorithms such as Structure2Vec perform k steps of local neighborhood aggregations, followed by an aggregation step. Note that the k steps of local neighborhood aggregations effectively compute a vector-representation for the vertices that is a function of their k-hop neighborhood subgraphs. Therefore, GCN algorithms are valid k-local-gather algorithms according to our definition above.
> > > > > >
> > > > > > *There are no inconsistencies in our definition.*
> > > > > >
> > > > > > We elaborate this point further below:
> > > > > >
> > > > > > Our definition of a k-local-gather algorithm does not place a restriction on the specific procedure that is used to compute the vertex representations in the local phase, or the output in the gather phase. The procedure used for computing the vertex representations in the local phase can be arbitrary, as long as it only depends on the k-hop neighborhood subgraph of vertices. Similarly, the procedure used for computing the output in the gather phase can be arbitrary, as long as it only depends on the set of vertex representations computed in the local phase.
> > > > > >
> > > > > > For example, a procedure in which the representation of a vertex is computed as the number of vertices in its k-hop neighborhood subgraph is a valid local phase procedure in a k-local-gather algorithm. Another example is a procedure where the representation of a vertex is 1 if there is a cycle in its k-hop neighborhood subgraph, and 0 if its k-hop neighborhood subgraph does not have cycle (i.e., is a tree).
> > > > > >
> > > > > > The k local steps of a GCN algorithm is also an example of a valid local phase procedure in a k-local-gather algorithm. To see this, we first explain a procedure that is easily seen to be a valid local phase procedure, and then point out that the procedure is equivalent to k step GCN algorithms.
> > > > > >
> > > > > > Consider a graph G whose vertices we want to represent. For a vertex v, we will first build a "computation tree" that depends on the k-hop neighborhood subgraph of v. Using this computation tree for v, we will obtain a representation for v.
> > > > > >
> > > > > > The computation tree will have v as the root. The root node has all the 1-hop neighbors of v (in G) as its children in the tree. Now, every node u at the first level of the tree has all the 1-hop neighbors of u as its children. We continue building the tree this way---at every level, adding the children in the 1-hop neighborhood of nodes of that level to the next level---till a depth of k. Notice that this computation tree depends only on the k-hop neighborhood subgraph of v.
> > > > > >
> > > > > > To compute the representation for v, we initialize the leaf nodes of v's computation tree with all-zero vectors. We will perform computation starting from the leaves of the tree and propagate vectors upwards towards the root of the tree. Each node, when it receives vectors from all of its children, computes a function over these vectors, and then forwards it to its parent in the computation tree. Finally, when the root receives vectors from all of its children, it computes a function over these vectors, and we obtain the representation for v.
> > > > > >
> > > > > > Therefore, a procedure that computes vertex representation this way (using a computation tree) is a valid local phase procedure in a k-local-gather algorithm.
> > > > > >
> > > > > > With a little bit of thought, it can be seen that this procedure corresponds precisely to how vertex representations are computed in a k-step message passing algorithm such as in Kipf and Welling. In a message passing algorithm, each vertex aggregates the current vector-state of its neighbors and updates its own vector-state based on that. In a k-step message passing algorithm, the update occurs k times. However, we will get the same resulting vertex representations if we "unfold" the computation tree around each vertex and propagate the vectors from the leaves to the root of this tree. The levels of the trees correspond to the message passing round number (with leaf nodes denoting round 0, the initial message passing state). As the computation progresses from the leaves to the root in the computation tree, the intermediate vector-outputs computed by nodes at level i correspond precisely to the vector-state of the vertices at the i-th round of message passing in G. Thus, a k-step message passing algorithm is a valid local phase procedure in a k-local-gather algorithm.

---

> > > > > > > ### Author Response · Authors · 2018-12-08
> > > > > > > **Author response 2**
> > > > > > >
> > > > > > > In summary, our definition of a k-local-gather algorithm consists of one local phase and one aggregation phase. Message passing algorithms such as Kipf and Welling consist of k steps of 1-hop local update operations. However, a k-step message passing algorithm such as Kipf and Welling is a valid k-local-gather algorithm where the k 1-hop local update operations *together* constitutes the local phase procedure of a k-local-gather algorithm.
> > > > > > >
> > > > > > > The 'k' in a k-local-gather algorithm stands for the size of the neighborhood considered, not the number of local steps. Our definition is consistent in saying GCN algorithms are valid k-local-gather algorithms.
> > > > > > >
> > > > > > > We have made every effort to clarify all the misunderstandings, in this thread and in the paper. We hope the reviewers will take that into account.

---

> > > > > > > > ### Comment · AnonReviewer2 · 2018-12-08
> > > > > > > > **I’m done**
> > > > > > > >
> > > > > > > > Dear authors,
> > > > > > > >
> > > > > > > > This is the last comment I’ll make. I do not appreciate that you make claims about there being no inconsistencies, write several paragraphs of arguments, none of which is resolving the issue.
> > > > > > > >
> > > > > > > > (1) “a k-step message passing algorithm such as Kipf and Welling is a valid k-local-gather algorithm where the k 1-hop local update operations *together* constitutes the local phase procedure of a k-local-gather algorithm.”
> > > > > > > >
> > > > > > > > (2) “The 'k' in a k-local-gather algorithm stands for the size of the neighborhood considered, not the number of local steps. Our definition is consistent in saying GCN algorithms are valid k-local-gather algorithms.”
> > > > > > > >
> > > > > > > > In (1) you write that k is the number of message passing steps.
> > > > > > > >
> > > > > > > > In (2) you write that k is the “size” of the neighborhood considered. (I assume size = distance.)
> > > > > > > >
> > > > > > > > The GCNs of Kipf and Welling do not only consider the 1-hop neighborhood, if the GCNs have more than one layer. Also, you state that GCNs are 4-local-gather. But that would mean there are only 4 local updates (according to (1)) which is not true.
> > > > > > > >
> > > > > > > > Your every effort to clarify misunderstandings currently makes things worse.
> > > > > > > >
> > > > > > > > The only way to make (1) and (2) consistent, is when you consider k to be the distance in the unrolled computation graph.
> > > > > > > >
> > > > > > > > But then GCNs (K&W) are not 4-local-gather.

---

> > > > > > > > > ### Author Response · Authors · 2018-12-09
> > > > > > > > > **Author response**
> > > > > > > > >
> > > > > > > > > Dear reviewer,
> > > > > > > > >
> > > > > > > > > First, we would like to again thank for taking time to provide feedback which has helped improve the paper. But with all due respect, we do not see inconsistencies in our definition or arguments. What we are saying is that a GCN with k convolutional layers uses a neighborhood of distance k around each vertex to compute the embedding for that vertex, and is therefore a k-local-gather algorithm by our definition. Please note that this is also in agreement with the Kipf and Welling paper (https://arxiv.org/pdf/1609.02907.pdf). On page 3, after defining graph convolution (Eq. (6)), the paper says:
> > > > > > > > >
> > > > > > > > > "Successive application of filters of this form then effectively convolve the k-th order neighborhood of a node, where k is the number of successive filtering operations or convolutional layers in the neural network model."
> > > > > > > > >
> > > > > > > > > Perhaps the confusion is that 4-local-gather in Proposition 2 refers to specific GCN realizations with up to 4 layers, such as Structure2Vec. We can clarify this in the paper by stating Proposition 2 in more general terms: A GCN with t layers is a t-local-gather algorithm.
> > > > > > > > >
> > > > > > > > > Please note that a GCN that has a fixed number of layers cannot propagate information across the entire graph (for large input graphs). The main point we are making is that the number of convolutions (layers) should vary dynamically based on the graph, and we provide a theoretical justification with Theorems 1 and 2.
> > > > > > > > >
> > > > > > > > > We reiterate that we have put in every effort into responding to reviewer comments thoroughly and respectfully. We hope the reviewer will extend us the same courtesy and consider our response.

---

### Official Review · AnonReviewer1 · 2018-11-03
**Need some clarification**

**Rating:** 5
**Confidence:** 5

**Review:**

This paper proposes a new representation learning model for graph optimization, Graph2Seq. The novelty of Graph2Seq lies in utilizing intermediate vector representation of vertices in the final representation. Theoretically, the authors show that an infinite sequence of such intermediate representations is much more powerful than existing models, which do not maintain intermediate representations. Experimentally, Graph2Seq results in greedy heuristics that generalize very well from small training graphs (e.g. 15 nodes) to large testing graphs (e.g. 3200 nodes).

Overall, the current version of the paper raises a number of crucial questions that I would like the authors to address before I make my decision.

First, some strengths of the paper:
- Theory: although I have not reviewed the proofs in details, the theorems are very interesting. If correct, the theorems provide a strong basis for Graph2Seq. In contrast, this aspect is missing from other work on ML for optimization.

- Experiments: the experiments are generally thorough and well-presented. The performance of Graph2Seq is remarkable, especially in terms of generalization to significantly larger graphs.

- Writing: the paper is very well-written and complex ideas are neatly articulated. I also liked the Appendix trying to interpret the trained model. Good job!

That being said, I have some serious concerns. Please clarify if I misunderstood anything and update the paper otherwise.

- Graph2Seq at test time: in section Testing, you explain how multiple solutions are output by G2S-RNN at intermediate "states" of the model, and the best w.r.t. the objective value is returned. If I understand all this correctly, you take the output of the T-th LSTM unit, run it through the Q-network, then select the next node (e.g. in a vertex cover solution). Then, the complexity should be O((E+V)*T_max*V), since the Graph2Seq operations are linear in the size of the graph O(E+V), a single G2S-RNN(i) takes O(V) times if you want to construct a cover of size O(V), and you repeat that process exactly T_max times, for each i between 1 and T_max. What's wrong in my understanding of G2S-RNN here? Or is your complexity incorrect?

- Local-Gather definition: in your definition of the Local-Gather model, do you assume that computations are performed for a single iteration, i.e. a single local step followed by a gather step? If so, then how is Graph2Seq infinity-local-gather? What does that even mean? I understand how some of the other GCNN-based models like Khalil et al.'s is 4-local-gather (assuming 4 embedding iterations of structure2vec), but how is Graph2Seq infinity-local-gather?

- Comparison to Structure2Vec: for fair comparison, why not apply Algorithm 2 to that method? Just run more embedding iterations up to T_max, and use the best among the solutions constructed between 1 and T_max.

Minor:
- Section 4: Vinyals et al. (2015) does not do any RL.

---

> ### Author Response · Authors · 2018-11-15
> **Author response**
>
> Thank you for the helpful comments.
>
> 1. Time complexity: Thank you for pointing out the complexity. It is as follows:
>
> (a) The time-complexity for one forward pass of G2S-RNN (e.g., to select one vertex in minimum vertex cover) is O((E + V)T_max). This is because during each step of Graph2Seq, O(E) operations are required to update the state at each vertex based on neighbors' states, and O(V) operations are required by the GRU to aggregate the states of all vertices (Equation 19 in appendix). Since these operations have to be repeated at each step, and there are at most T_max steps, the time-complexity is O((E + V)T_max).
>
> (b) For a fixed number of steps T, the time-complexity to compute a complete solution (e.g., to select multiple vertices such that they form a valid vertex cover) is O((E + V)T_max * V). This is because selecting one vertex has complexity O((E + V)T_max), and we may have to select O(V) vertices to obtain a valid solution to the input graph.
>
> (c) The overall time-complexity is O((E + V)T_max * V * T_max). This is because the final solution is computed by first computing valid solutions for each T=1,2,..,T_max, and then picking the best valid solution from among them. Computing a valid solution for a fixed T takes O((E + V)T_max * V) as mentioned above, and we have to repeat the process T_max times.
>
> Note that aggregating states from all the vertices in the GRU is a hyperparameter choice. If only local neighborhood states are used in the GRU, the time-complexity in step (a) above becomes O(ET_max + V).
>
> We will clarify the complexity in Section 5.
>
> 2. Local Gather definition: Yes, the definition of local-gather consists of one local step followed by one gather step. An algorithm is k-local-gather if in the local step, each vertex computes an embedding based on the k-hop neighborhood graph around the vertex. Structure2Vec is 4-local-gather because the four embedding iterations cause each node's embedding to depend on its 4-hop neighborhood. Graph2Seq is infinity-local-gather since the infinite number of embedding iterations cause each node's embedding to depend on the entire graph---not just on vertices a constant number of hops away from the node. Infinity is used to emphasize that the local graph neighborhoods on which the node embeddings depend are not constrained in size. For a specific graph G with diameter dia(G), it is also true that Graph2Seq is dia(G)-local-gather. We will include a remark to explain that infinity-local-gather means that a node's embedding can depend on the entire graph, regardless of the graph size.
>
> 3. Comparison to Structure2Vec: We will include an experiment that applies algorithm 2 to Structure2Vec in the paper. Note, however, that applying this procedure to Structure2Vec implicitly uses the sequence of Structure2Vec embeddings as the embedding for each vertex. Therefore, this method is a different instantiation of our idea of using sequences for node embeddings. In particular, like Graph2Seq, this method will also consider neighborhoods of different sizes around each vertex for different graphs. The only difference is that Graph2Seq additionally uses an LSTM to process the sequence. Therefore, we indeed expect that the combination of Algorithm 2 with Structure2Vec will also perform well.

---

> > ### Comment · AnonReviewer1 · 2018-12-04
> > **Thank you**
> >
> > Thanks for the response and the updated paper.
> >
> > Regarding point 1, the explanation in the appendix is now clear; thank you.
> > Regarding point 2, I posted a comment in reply to the discussion with Reviewer 2 below.
> > Regarding point 3, it's great that you managed to run all these additional experiments so quickly. As you mention in the paper, S2V-Dynamic seems to perform similarly to Graph2Seq.
> >
> > At the moment, I am most comfortable keeping my score unchanged. The theory and experiments seem to suggest that you have an interesting model but it is not clear that it is better than similar existing models (e.g. S2V-Dynamic) nor theoretically more powerful (ref. discussion about the k-local-gather notion below).

---

> > > ### Author Response · Authors · 2018-12-08
> > > **Author response**
> > >
> > > Thank you for your feedback.
> > >
> > > Please note that S2V-dynamic is not an existing model. We are not aware of any prior work that has proposed varying the number of local steps in GCNs dynamically.
> > >
> > > Also, the point of our work is not the specific architectural differences between S2V-dynamic and G2S-RNN. In fact, our method can be used on top of *any* GCN method to improve generalization.
> > >
> > > Rather, our point is that varying the number of local steps used in GCN algorithms results in better generalization. Therefore, the fact that S2V-dynamic has better performance than standard S2V (with a fixed number of iterations) affirms our main result.
> > >
> > > Furthermore, looking at the sequence of local step outputs results in even better generalization. This is why using a GRU in G2S-RNN improves performance even over S2V-dynamic.

---

### Official Review · AnonReviewer3 · 2018-11-05
**A good paper for the conference**

**Rating:** 6
**Confidence:** 3

**Review:**

Graph representation techniques are important as various applications require learning over graph-structured data. The authors proposed a novel method to embedding a graph as a vector. Compared to Graph Convolutions Neural Networks (GCNN), the proposed are able to handle directed graphs while GCNN can not. Overall the paper is good, the derivation and theory are solid. The authors managed to prove the proposed representation is somehow lossless, which is very nice. The experiment is also convincing.

My only concern is as follows. The authors claim that Eq. (1) is able to handle features on vertices or edges. However, in the current formulation, the evolution only depends on vertex features, thus how can it handle edge features?

---

> ### Author Response · Authors · 2018-11-15
> **Author response**
>
> We thank the reviewer for the helpful comments.
>
> Including edge features: There are a few different ways to include edge features. One way would be to include a second term $\sum_{e\in\eta(v)}y_e(t)$, where $\eta(v)$ are edges incident to node v and y_e are edge features of edge e, inside the ReLU function of Equation 1. Another way is to transform the graph with edge features into a new (larger) graph where there are no edge features. This is done by converting the original graph into a new bipartite graph where one partite corresponds to vertices of the original graph, and the other partite corresponds to edges of the original graph. Each edge-node in the bipartite graph is connected to the two vertex-nodes that constitute its end points in the original graph. The edge-nodes have the edge features of the original graph, while the vertex-nodes have the vertex features. We will explain this in the revision.

---

### Author Response · Authors · 2018-11-24
**Revised paper**

We have uploaded a revision of the paper with the following changes:

1. We have run experiments where the depth of Structure2Vec is varied as in Algorithm 2, and have included its results in Figures 2, 3 and 4. This algorithm is called S2V-dynamic in the Figures and can be found as the black curve.

2. We have collected run times of all the schemes for computing minimum vertex cover on Erdos-Renyi graphs of various sizes. Appendix D.3 presents these results. We are currently collecting the run times for max cut, maximum independent set and will include those results also shortly.

3. We have expanded the proof of Theorem 2 in Appendix B.5 to clarify the extension of proof to the general k case. We have added a subfigure in Figure 5 showing how the trees can be constructed for k=3. A new Figure 6 has also been added showing the optimum vertex covers for the trees of Figure 5. Lastly, we have included a remark after the proof of Theorem 2 explaining the effect of node labels on local-gather algorithms.

4. We have modified the introduction to better motivate why the scalability of Graph2Seq is useful. Specifically, we have included device placement in TensorFlow, query optimization and job scheduling as examples of practical applications where learning scalable graph algorithms, such as Graph2Seq, could be beneficial.

5. We have explained what infinity-local-gather means in Section 3.2.

6. We have briefly explained how node labels can influence the number of local steps after Theorem 2 in Section 3.2 (detailed explanation provided in Appendix B.5).

7. We have provided an explanation on how edge features can be included in Graph2Seq in Appendix B.1.

8. We have mentioned the time-complexity of G2S-RNN in Section 5 (under Testing subsection) and have derived it in Appendix C.3.

We once again thank the reviewers for their valuable feedback which has helped greatly improve the paper.

---

> ### Author Response · Authors · 2018-11-26
> **Revised paper**
>
> We have uploaded a revision including the run times for max cut and maximum independent set problems, in Appendix D.3.

---

### Public Comment · ~Abishek_Sankararaman1 · 2018-12-27
**A Simpler Proof of Theorem 2 ?**

Fix any $k \in \mathbb{N}$ and consider two graphs, one is a cycle consisting of 4k nodes and the other is two disjoint cycles of 2k nodes each. Consider the function f(G) to denote the number of connected components. Clearly, the k hop neighborhood of all nodes in both the graph are identical, while the number of connected components are different.

Does this argument also prove Theorem 2 ? Or did I misunderstand some thing in the definition and the proofs ?

---

> ### Author Response · Authors · 2018-12-29
> **response**
>
> Thanks for the comment. Yes the proof you have mentioned is valid, and useful in its conciseness.
>
> However, the proof we have given in the paper shows a construction for a “simpler” case where the graph is not disconnected. This is because in many practical applications if the input graph is disconnected one may do some preprocessing and either (i) process the individual connected components separately, or (ii) add extra edges (i.e., edges with special features) and make the graph connected.
>
> Since the paper focuses on combinatorial optimization, we have used minimum vertex cover as the function of choice in our proof. However, other functions or graph constructions may be possible even in the case where the graphs are not disconnected.

---

### Meta-Review · Area_Chair1 · 2018-12-14
**Good paper, but there are some issues with the theory (either correctness or clarity) that need to be resolved.**

**Confidence:** 3
**Recommendation:** Reject

**Metareview:**

This was an extremely difficult case. There are many positive aspects of Graph2Seq, as detailed by all of the reviewers, however two of the reviewers have issue with the current theory, specifically the definition of k-local-gather and its relation to existing models. The authors  and reviewers have had a detailed and discussion on the issue, however we do not seem to have come to a resolution. I will not wade into the specifics of the argument, however, ultimately, the onus is on the authors to convince the reviewers of the merits/correctness, and in this case two reviewers had the same issue, and their concerns have not been resolved. The best advice I can give is to consider the discussion so far and why this misunderstanding occurred, so that it might lead the best version of this paper possible.